



# Long-term Regional Trends of Nitrogen and Sulfur Deposition in the United States from 2002 to 2017

Sarah E. Benish[1], Jesse O. Bash[2], Kristen M. Foley[2], K. Wyat Appel[2], Christian Hogrefe[2], Rob Gilliam[2], George Pouliot[2]

[1] Oak Ridge Institute for Science and Education (ORISE), US Environmental Protection Agency, Research Triangle Park, NC 27711, USA

[2] US Environmental Protection Agency, Research Triangle Park, NC 27711, USA

*Correspondence to*: Sarah E. Benish (benish.sarah@epa.gov) and Jesse O. Bash (bash.jesse@epa.gov)

**Abstract.** Atmospheric deposition of nitrogen (N) and sulfur (S) compounds from human activity has greatly declined in the United States (US) over the past several decades in response to emission controls set by the Clean Air Act. While many studies have investigated the spatial and temporal trends of atmospheric deposition, few assess dry deposition, incorporate a measurement-model fusion approach to improve wet deposition estimates, or focus on changes within specific US climate regions. In this analysis, we evaluate wet, dry, and total N and S deposition from multiyear simulations across climatologically consistent regions within the contiguous US (CONUS). Community Multiscale Air Quality (CMAQ) model estimates from 2002 to 2017 from the EPA's Air QUALity TimE Series (EQUATES) project contain important model updates to atmospheric deposition algorithms compared to previous model data, including the new Surface Tiled Aerosol and Gaseous Exchange (STAGE) bidirectional deposition model and improvements to organic N chemistry. The model generally underestimates wet deposition of $SO_4$, $NO_3$, and $NH_4$ compared to National Atmospheric Deposition Program observations. Measurement-model fusion employing a precipitation- and bias- correction to wet deposition estimates is found to reduce model bias and improve correlations compared to the unadjusted model values. Comparisons to Clean Air Status and Trends network ambient concentrations show the model underestimates $NH_4$ and $SO_4$ and overestimates $SO_2$ and $TNO_3$. Model agreement is poor over parts of the West and Northern Rockies, due to errors in precipitation estimates caused by complex terrain and uncertainty in emissions at the relatively coarse 12 km grid resolution used in this study. Total deposition of N and S in the eastern US is larger than the western US with a steeper decreasing trend from 2002-2017, i.e., total N declined at a rate of approximately −0.30 kg-N/ha/yr in the Northeast and Southeast and by −0.06 kg-N/ha/yr in the Northwest and Southwest. Widespread increases in reduced N deposition across the Upper Midwest, Northern Rockies, and West indicate evolving atmospheric composition due to increased precipitation amounts over some areas, growing agricultural emissions, and regional $NO_x/SO_x$ emission reductions; these increases in reduced N deposition are generally masked by the larger decreasing oxidized N trend. We find larger average declining trends of total N and S between 2002-2009 than 2010-2017, suggesting a slowdown of the rate of decline. The average total N deposition budget over the CONUS decreases from 7.8 kg-N/ha in 2002 to 6.3 kg-N/ha in 2017 due to declines in oxidized N deposition from $NO_x$ emission controls. Across the US during the 2002-2017 time period, the average contribution of dry deposition to the total N





deposition budget drops from 60% to 52%, whereas wet deposition dominates the S budget rising from 45% to 68%. Our analysis extends upon the literature documenting the growing contribution of reduced N to the total deposition budget, particularly in the Upper Midwest and Northern Rockies, and documents a slowdown of the declining oxidized N deposition trend, which may have consequences on vegetation diversity and productivity. Future progress in decreasing the total N budget remains exceedingly difficult without new controls on ammonia emissions, including from agricultural sources as the demand for food grows and oxidized N emissions continue to decrease due to emission controls implemented to achieve the National Ambient Air Quality Standards.

## 1 Introduction

Human activity doubled the amount of reactive nitrogen (Nr) in the environmental globally over the past century (Fowler et al., 2013). Major sources responsible for the increase include fossil fuel combustion from vehicles and electric utilities emitting nitrogen oxides ($NO_x=NO+NO_2$) and sulfur dioxide ($SO_2$), and agricultural activities releasing ammonia ($NH_3$) (Galloway et al., 2008). After entering the atmosphere, the major nitrogen (N) and sulfur (S) removal pathways occur by precipitation (wet deposition) or uptake by terrestrial and aquatic vegetation (dry deposition). Consequences of deposition include risks to public health (Townsend et al., 2003) and damages to ecosystems, such as acidification of soil and waterways, changes to species composition (Bobbink et al., 2010), and massive cyanobacteria and algal blooms (Jaworski, 1990), which may cause physiological stress and compromise immune function among wildlife (Refsnider et al., 2021). Despite adverse effects of deposition to humans and the environment, the total (wet+dry) N and S budgets remain uncertain, particularly for sensitive ecosystems in critical regions due to observational gaps.

Deposition monitoring and assessment played a decisive role informing the United States (US) Clean Air Act Amendments (CAAA) of 1990 (United States Congress, 1990). Wet deposition, sampled in rain or snow, is measured by the National Atmospheric Deposition Program (NADP) National Trends Network (NTN), collecting samples weekly since 1978. Dry deposition measurements are inferred by combining weekly measured concentrations from the Clean Air Status and Trends Network (CASTNET) and modeled deposition velocities from the Multilayer Model (MLM) (Meyers et al., 1998). Despite providing critical deposition information, the limited number of NADP and CASTNET sites in essential locations, such as areas with complex terrain, high elevation, or in forest ecosystems, restrict a thorough understanding on the amount and consequences of deposition. Additionally, networks are unable to capture the full budget of Nr. For instance, organic N species are estimated to contribute ~25% to total Nr (Jickells et al., 2013), but are not easily quantified by the NTN due to limitations in field and laboratory techniques (Walker et al., 2012).

Chemical transport models (CTMs) can be used to study deposition relationships and trends for locations without measurements and compounds that are challenging to quantify (e.g., organic N). However, to provide reliable estimates,



CTMs estimates must first be evaluated with measurement data. Various studies have compared N and S wet deposition and concentrations estimated by CMAQ with observed values in the US (Xing et al., 2015; Zhang et al., 2018; Appel et al., 2011), providing a wide range of agreement with measurements. Zhang et al. (2018) found adjusting CMAQv5.0.2 wet deposition estimates with observed precipitation results in a larger negative normalized mean bias (NMB) than the original CMAQ estimates for $TNO_3$ (−31.6% unadjusted vs −35.6% adjusted), $NH_x$ (−30.9% unadjusted vs −35.1% adjusted), and

total S (−5.1% unadjusted vs −10.5% adjusted). Similar model agreement were published using CMAQv4.7 (Appel et al., 2011) and attributed to coarse grid resolution (36 km). In Europe, Theobald et al. (2019) characterized the model bias of wet deposition of $NO_x$, $NH_x$, and $SO_x$ from six CTMs between 1990 and 2010 to range from 30-40%. Biased model simulations may lead to incorrect scientific conclusions, such as critical loads designations (Williams et al., 2017). Methods to fuse model estimates with measured values can therefore be used to provide more reliable insights into "biogeochemical cycles

and assessments of ecosystems and human health effects" (World Meteorological Organization, 2017). As such, the NADP Total Deposition Science Committee (TDEP, see www.nadp.slh.wisc.edu/committees/tdep) advances methods to improve estimates of atmospheric deposition (see Schwede and Lear (2014)), although no bias-correction for wet deposition is currently employed in products available to the public.

In this study, we assess deposition from the EPA's Air QUAlity TimE Series (EQUATES) project (https://www.epa.gov/cmaq/equates), which utilized the Community Multiscale Air Quality (CMAQ; https://www.epa.gov/cmaq) version 5.3.2 modeling platform. The EQUATES project used consistent methodology (e.g., input data, processing methods, model versions) with deposition science updates compared to previous model versions, including the new Surface Tiled Aerosol and Gaseous Exchange (STAGE) bidirectional deposition model and improvements

to organic N chemistry. Long-term air quality simulations like EQUATES are used by the scientific community in ecological and epidemiological assessments, including critical loads analyses. By employing these science updates in the model, consistent methodology, finer grid spacing (12 km), and a bias-adjustment to wet deposition, we evaluate our best understanding of simulated deposition trends in the US. We evaluate the model agreement with observations of wet deposition and concentrations used to infer dry deposition and describe a measurement-model fusion technique in Section 2.

In Section 3, we focus on the modeled trend and budget of N and S deposition over nine climatologically consistent regions within the contiguous US (CONUS) over a period of large emission reductions designated by the CAAA. Section 4 ends with Conclusions.

## 2 Methods and Materials

### 2.1 EQUATES Model Configuration

The long-term EQUATES simulations from 2002-2017 were run using the CMAQ  model version 5.3.2. A summary of recent CMAQ version 5.3 model updates and their impact on modeled concentrations is provided in Appel et al. (2021).

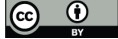



Modeled data cover the CONUS using 12 km grid spacing and the Northern Hemisphere using 108 km grid spacing. The Weather Research and Forecasting version 4.1.1 (WRFv4.1.1) was used for meteorology. Lateral boundary conditions for the CONUS domain were provided by Northern Hemispheric simulation. Hemispheric and North American emission

inventories were specifically prepared for EQUATES to ensure consistent input data and methods across all years. The Surface Tiled Aerosol and Gaseous Deposition (STAGE) option in CMAQ v5.3.2 (Galmarini et al., 2021) was used to estimate atmospheric dry deposition rates. The STAGE dry deposition option generally performs similar to M3Dry (the other dry deposition option in CMAQ v5.3.2) when compared to network observations of ambient gaseous and aerosol pollutant concentrations (Appel et al., 2021) while providing additional land use specific deposition data useful for assessments of

ecosystem exposure (Hood et al., 2021).

## 2.2 US Deposition Observations

We assess CMAQ's ability to reproduce wet deposition and ambient concentrations of N and S species measured by the NADP NTN and EPA's CASTNET networks (Figure 1). Since anthropogenic sources dominate deposition and their area of influence is  largely regional (Paulot et al., 2013), we evaluate the agreement over CONUS climate regions from 2002-2017

following the NOAA definition defined by Karl and Koss (1984), although these groupings do not imply the regions are isolated from each other. The nine climate regions include the Northwest, West, Northern Rockies, Southwest, South, Ohio Valley, Southeast, and Northeast as shown in Figure 1.

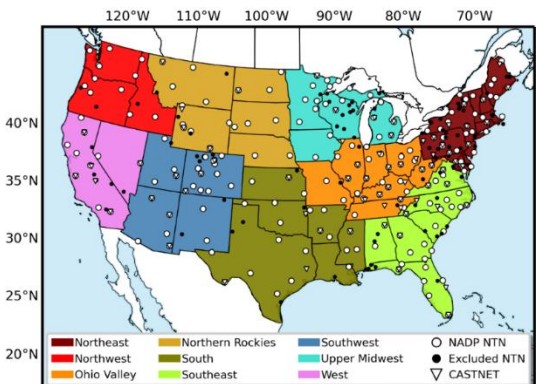

**Figure 1. Site locations of the 200 National Atmospheric Deposition Program (NADP) National Trends Network (NTN, circles) and**
**75 Clean Air Status and Trends Network (CASTNET, triangles) examined in this study. Color-coded US climate regions shown in this map are referred to throughout this analysis. The black-bordered white circles indicate NADP NTN sites that meet annual completeness criteria for 13 years of the timeseries and examined in the model evaluation presented in Section 3.1.**

The NADP's NTN (http://nadp.slh.wisc.edu) currently collects weekly precipitation samples at 263 locations across the US. Samples are analyzed by ion chromatography or flow injection analysis to quantify wet deposition of nitrate (NO$_3$),

ammonium (NH$_4$), and sulfate (SO$_4$), as well as several other compounds. First, modeled wet deposition of NO$_3$, NH$_4$, and





$SO_4$ is calculated using the approach by Appel et al. (2011) that accounts for chemical transformations of several species in the aqueous phase. Then, the model and observations are paired in time and space. Following similar completeness criteria thresholds as in Zhang et al. (2019), NTN sites must be available for at least 13 of the 16 years simulated and have at least 60% annual coverage each year to be considered for analysis. Applying these criteria, 200 NTN sites meet the minimum requirements for our analysis (Table S1). The CASTNET measures concentrations and estimates dry deposition at 99 sites in the US (https://www.epa.gov/castnet). Weekly ambient concentrations of gases and particles are collected with an open-face 3-stage filter pack. Since the reported dry deposition values estimated by CASTNET are model derived, we instead evaluate the CMAQ performance simulating the ambient air concentrations of sulfur dioxide ($SO_2$), sulfate ($SO_4$), total nitrate ($TNO_3=NO_3+HNO_3$), and ammonium ($NH_4$). CASTNET sites available for at least 13 of the 16 years simulated with 75% annual coverage each year are included in this analysis, resulting in 75 valid sites meeting the minimum requirements (Table S2). The EQUATES simulations are evaluated by comparing with available monitoring data using several model performance statistics (Simon et al., 2012).

**2.3 Measurement-Model Fusion Technique**

The modeled wet deposition fields are adjusted to account for input biases and uncertainty in the chemical and physical processes governing deposition. We apply a measurement-model fusion technique previously described by Zhang et al. (2019) to adjust the modeled wet deposition fields of inorganic N ($NO_3+NH_4$) and S, briefly described here. First, the modeled wet deposition fields from EQUATES are adjusted using observation-based 4 km precipitation fields generated by the Parameter-elevation Regressions on Independent Slopes Model (PRISM, https://prism.oregonstate.edu/), annually accumulated and regridded to the 12 km CMAQ domain (EQUATES$_{precip-adj}$). Since PRISM incorporates observed elevation and climatic factors to estimate precipitation, NTN and TDEP maps also used these data to improve estimates of wet deposition over complex terrain  (Schwede and Lear, 2014). In each grid cell, the EQUATES modeled annual total wet deposition (WD$_{mod}$) is adjusted by the ratio of annually accumulated precipitation from PRISM (Precip$_{obs}$) to the WRF estimated annual accumulated rainfall (Precip$_{mod}$) (Equation 1):

$$EQUATES_{precip-adj} = \frac{Precip_{obs}}{Precip_{mod}} \times WD_{mod} \qquad (1)$$

After adjusting simulated wet deposition by precipitation, an additional bias-adjustment (EQUATES$_{bias-adj}$) is applied using all NTN observations that meet annual data completeness. For each NTN site, the median ratio between the EQUATES$_{precip-adj}$ wet deposition and the NTN observations is calculated for all sites within a 300 km radius. Then, the median biases are interpolated using universal kriging with a linear trend in the spatial coordinates and an exponential covariance structure using python's pykrige module (https://github.com/GeoStat-Framework/PyKrige). A cross-validation analysis employing this measurement model fusion technique with CMAQv5.0.2 is presented in Zhang et al. (2019). The final wet deposition





bias-adjustment (Equation 2) is computed by multiplying the precipitation-adjusted wet deposition by the inverse of the bias field, $b$:

$$EQUATES_{bias-adj} = \frac{1}{b} \times EQUATES_{precip-adj} \tag{2}$$

**3 Results and Discussion**

**3.1 Model Performance**

In this section, we investigate the EQUATES model performance estimating wet deposition and ambient concentrations of N and S compounds throughout the US. The impact of the precipitation and bias correction technique on reproducing 2002-2017 accumulated measured wet deposition of $NH_4$, $NO_3$ and $SO_4$ is shown in the Taylor Diagram in Figure 2 and
summarized in Table 1. In addition, since previous modeling studies demonstrated moderate skill simulating wet deposition (Appel et al., 2011; Zhang et al., 2018), we also compare to an earlier timeseries utilizing CMAQv5.0.2 (ECODEP, 2002-2012 only). Little difference was found between EQUATES and ECODEP simulated wet deposition statistics over the CONUS, although the improvement in the EQUATES estimates is more noticeable over regions with improved precipitation estimates. For example, a high precipitation bias in the WRFv3.4 meteorology used for ECODEP is reduced or eliminated in
the WRFv4.1.1 meteorology used for EQUATES. The decrease in bias is likely attributed to the implementation of a hybrid coordinate system in WRFv4.1.1 which has been found to improve precipitation estimates at high elevation sites in the western US (Beck et al., 2020), although the EQUATES precipitation is still biased low on average relative to PRISM. The close grouping of markers in Figure 2a compared to the other species presented in Figure 2 reflect the smaller standard deviation in the $NH_4$ wet deposition measurements (1.5 kg/ha), which are comparable and slightly underestimated among the
various models and adjustments. Applying the precipitation-adjustment ($EQUATES_{precip-adj}$) reduces the underestimation of $NH_4$ and $SO_4$ wet deposition and increases the overestimation of $NO_3$ wet deposition compared to the unadjusted model output (Table 1). The bias-adjusted $NH_4$, $NO_3$, and $SO_4$ wet deposition estimates have a similar standard deviation as the observations (falling near the dashed arc), and also feature a lower root-mean square error (RMSE) (dotted semicircles) and higher correlation compared to the other estimates. Interannually, the bias-correction to EQUATES output reduces the NMB
of $NH_4$ and $SO_4$ wet deposition by 10-30% and 5-20%, respectively, compared to the unadjusted model output, with the largest improvement occuring from 2010-2017 (Figure S1). Corrections applied to modeled $NO_3$ wet deposition reduce the NMB by ~20% in the beginning of the timeseries (2002-2009) and <10% at the end (2010-2017), possibly related to more uncertainty in emissions earlier in the earlier time period. The bias-corrections to $NH_4$ wet deposition are particularly noteworthy in the Upper Midwest, Northern Rockies, Ohio Valley, and South (Figure S2), improving the agreement between
the modeled and observed standard deviation, increasing the correlation, and lowering the RMSE compared to the unadjusted model values. The bias-adjustment greatly improves the agreement of $NO_3$ and $SO_4$ wet deposition to observations in the Northeast, Northwest, and Ohio Valley by similarly improving the model performance metrics (Figures





S3 and S4). Due to the large improvements in estimating wet deposition using the measurement model fusion approach, we use these corrections when assessing wet and total deposition throughout the rest of the analysis.

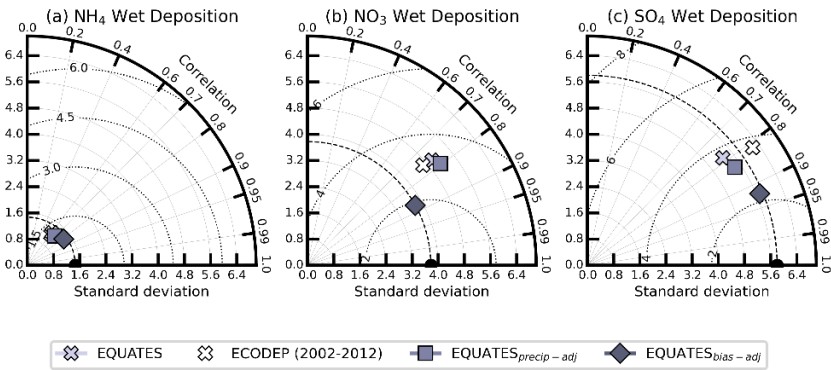

**Figure 2. Taylor plot comparing annual accumulated wet deposition (kg/ha) of NH₄ (a), NO₃ (b), and SO₄ (c) collected at NTN sites (black circles) with model output. The symbols differentiate between the various models (EQUATES and ECODEP) and wet deposition corrections as described in the text. The azimuthal angle denotes the Pearson correlation coefficient ($r^2$), the dashed radial distance shows the standard deviation (kg/ha), and the dotted semicircles centered at the observation marker (black circle) denotes the root-mean-square error.**

**Table 1. EQUATES model performance metrics of annual (2002-2017) accumulated wet deposition of NH₄, NO₃, and SO₄, including the Pearson correlation coefficient ($r^2$), mean bias (MB, kg/ha), and normalized mean bias (NMB, %).**

| Metric | Wet Deposition Correction | NH₄ | NO₃ | SO₄ |
|---|---|---|---|---|
| $r^2$ | No adjustment | 0.61 | 0.77 | 0.78 |
| | Precipitation adjustment | 0.68 | 0.79 | 0.83 |
| | Bias adjustment | 0.81 | 0.88 | 0.92 |
| Mean Bias (MB, kg/ha) | No adjustment | −0.49 | 0.66 | −0.92 |
| | Precipitation adjustment | −0.37 | 1.18 | −0.43 |
| | Bias adjustment | −0.07 | −0.03 | −0.10 |
| Normalized Mean Bias (NMB, %) | No adjustment | −19.9 | 9.64 | −12.2 |
| | Precipitation adjustment | −15.1 | 17.1 | −5.73 |
| | Bias adjustment | −2.83 | −0.48 | −1.31 |

Figure 3 provides an overall comparison between the bias-corrected EQUATES wet deposition and NTN observations, along with an evaluation of simulated and observed precipitation. Table 2 provides summary statistics of wet deposition and precipitation for each climate region. A slight negative bias of modeled precipitation and bias-corrected NH₄, NO₃, and SO₄





wet deposition is found for the entire CONUS, although agreement across the climate regions varies widely. The model generally exhibits better performance in the eastern climate regions, especially the Ohio Valley and Upper Midwest, compared to the rest of the CONUS. Agreement in the West, Northwest, and Northern Rockies is particularly poor, likely due to the errors simulating precipitation in the complex terrain (Table 2). In these western climate regions, precipitation is
generally overestimated, consistent with previous model versions (Zhang et al., 2018).

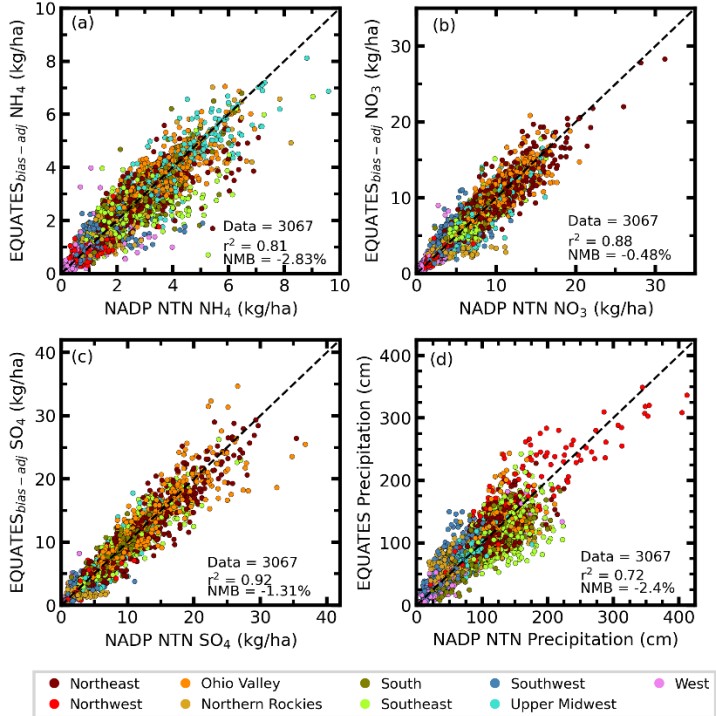

**Figure 3. Scatter plots of annual accumulated bias-adjusted modeled and NTN observed wet deposition (kg/ha) of ammonium (a, NH$_4$), nitrate (b, NO$_3$), and sulfate (c, SO$_4$) from 2002 to 2017 colored by the climate region. Panel d shows observed and modeled**
**precipitation (cm).**

The 16-year total NH$_4$ wet deposition is highest in the Upper Midwest, seen in both the model and measurements, and lowest in the Northwest. The model generally underestimates NH$_4$ wet deposition for all regions except the Ohio Valley, with MB values ranging from 0.03 kg/ha in the Ohio Valley to −0.15 kg/ha in the Southeast. The correlation coefficients for wet NH$_4$ deposition are on average stronger in the northern ($r^2>0.66$) than southern US ($r^2<0.60$). The 16-year total NO$_3$ wet
deposition is also highest in the Upper Midwest, but lowest in the West. The simulation overestimates NO$_3$ wet deposition in the West, South, Ohio Valley, and Southwest with MB values all approximately 0.10 kg/ha, while all other regions are underestimated with a MB ranging from −0.01 kg/ha in the Northwest to −0.39 kg/ha in the Northern Rockies. Correlation




coefficients for $NO_3$ wet deposition are greater than 0.60 in all regions except the Southwest and Northern Rockies ($r^2$~0.50).
Similar to $NO_3$ wet deposition, the 16-year total wet $SO_4$ deposition is also highest in the Upper Midwest and lowest in the

West. The simulation overestimates $SO_4$ wet deposition across most of the southern US, where the MB ranges from 0.01 kg/ha (Southwest) to 0.16 kg/ha (West). Most other regions show an underestimation of $SO_4$ wet deposition with MB ranging from −0.03 kg/ha (Ohio Valley) to −0.32 kg/ha (Northeast). Correlation coefficients are strongest for $SO_4$ wet deposition than the other species, particularly in the northern US where $r^2$ values are generally greater than 0.80. Like $NH_4$ and $NO_3$ wet deposition, correlation coefficients for $SO_4$ wet deposition are weakest in the Southwest and West. The model

also captures the clear downward trends of $NO_3$ and $SO_4$ wet deposition ($r^2$=0.79 and 0.94, respectively), although the magnitudes of the decreasing modeled trends are slightly underestimated (Figure S5). The weaker correlation between the measured and modeled trends of $NH_4$ wet deposition ($r^2$=0.43) is impacted by the smaller magnitude of the observed and modeled $NH_4$ trends and reflects moderate model performance reproducing decreasing trends in many parts of the southern US and increasing trends in the northern US.


**Table 2. Evaluation of bias-adjusted $SO_4$, $NO_3$, and $NH_4$ wet deposition and precipitation compared to NTN measurements in the contiguous US and nine climate regions that meet the completeness criteria.**

| Climate Region | Sites (N) | SO₄ | | | | NO₃ | | | | NH₄ | | | | Precipitation | | | |
|---|---|---|---|---|---|---|---|---|---|---|---|---|---|---|---|---|---|
| | | Mean Obs/Mod (kg/ha) | MB (kg/ha) | NMB (%) | r² | Mean Obs/Mod (kg/ha) | MB (kg/ha) | NMB (%) | r² | Mean Obs/Mod (kg/ha) | MB (kg/ha) | NMB (%) | r² | Mean Obs/Mod (cm) | MB (cm) | NMB (%) | r² |
| United States | 200 (3067) | 7.50/7.40 | −0.10 | −1.31 | 0.92 | 6.90/6.87 | −0.03 | −0.48 | 0.88 | 2.43/2.36 | −0.07 | −2.83 | 0.81 | 96.7/94.4 | −2.32 | −2.40 | 0.72 |
| Northeast | 35 (531) | 11.48/11.16 | −0.32 | −2.78 | 0.89 | 9.87/9.79 | −0.08 | −0.80 | 0.82 | 2.64/2.56 | −0.08 | −3.06 | 0.68 | 122.1/116.1 | −6.05 | −4.95 | 0.27 |
| Northwest | 12 (184) | 2.53/2.45 | −0.08 | −3.26 | 0.81 | 2.51/2.50 | −0.01 | −0.39 | 0.82 | 0.82/0.73 | −0.09 | −10.6 | 0.52 | 124.5/136.5 | 12.0 | 9.62 | 0.87 |
| Ohio Valley | 27 (418) | 13.1/13.1 | −0.03 | −0.25 | 0.82 | 10.33/10.44 | 0.11 | 1.02 | 0.67 | 3.57/3.59 | 0.03 | 0.76 | 0.66 | 116.9/118.2 | 1.32 | 1.13 | 0.36 |
| Northern Rockie | 19 (284) | 2.71/2.48 | −0.23 | −8.37 | 0.61 | 4.07/3.68 | −0.39 | −9.49 | 0.54 | 2.13/2.02 | −0.11 | −5.17 | 0.89 | 57.4/64.8 | 7.40 | 12.9 | 0.42 |



| s | | | | | | | | | | | | | | | | | |
|---|---|---|---|---|---|---|---|---|---|---|---|---|---|---|---|---|---|
| **South** | 21 (323) | 7.82/7.89 | 0.07 | 0.87 | 0.87 | 6.91/7.01 | 0.09 | 1.37 | 0.86 | 2.82/2.74 | −0.08 | −2.69 | 0.64 | 97.0/82.6 | −14.4 | −14.9 | 0.76 |
| **Southeast** | 29 (447) | 9.82/9.70 | −0.11 | −1.16 | 0.84 | 7.53/7.43 | −0.10 | −1.29 | 0.69 | 2.31/2.16 | −0.15 | −6.32 | 0.59 | 128.7/111.0 | −17.7 | −13.7 | 0.28 |
| **Southwest** | 26 (400) | 2.22/2.23 | 0.01 | 0.46 | 0.47 | 3.70/3.86 | 0.16 | 4.27 | 0.52 | 1.20/1.16 | −0.04 | −2.93 | 0.45 | 49.2/58.7 | 9.45 | 19.2 | 0.65 |
| **Upper Midwest** | 21 (330) | 6.65/6.53 | −0.11 | −1.71 | 0.86 | 8.00/7.83 | −0.18 | −2.24 | 0.75 | 3.82/3.76 | −0.06 | −1.45 | 0.77 | 82.0/83.2 | 1.22 | 1.49 | 0.50 |
| **West** | 10 (150) | 1.16/1.32 | 0.16 | 14.1 | 0.56 | 1.80/1.91 | 0.11 | 6.01 | 0.60 | 0.87/0.81 | −0.07 | −7.72 | 0.34 | 53.2/50.6 | −2.64 | −4.97 | 0.81 |

To indirectly evaluate the ability of the EQUATES simulations to reproduce dry deposition, we compare the model

simulated annual average concentrations of $SO_2$, $SO_4$, $TNO_3$, and $NH_4$ with observations collected at CASTNET sites in

Figure 4 and in Table 3. Correlations between measured and modeled mean concentrations show good agreement for $SO_2$

($r^2$=0.90), $SO_4$ ($r^2$=0.96), $TNO_3$ ($r^2$=0.81), and $NH_4$ ($r^2$=0.94). Overall, the model shows positive biases for $TNO_3$ and $SO_2$

concentrations (2.8% and 26.0% respectively) and negatives biases for $SO_4$ and $NH_4$ concentrations (−15.4% and −17.4%

respectively), but generally improved from CMAQv5.0 (Xing et al., 2015).  The highest 16-year average concentrations for

all species are found in the Upper Midwest for both the simulations and observations. Average modeled $TNO_3$ concentrations

are mostly overestimated in the northern US, with MB ranging from 0.02 μg/m$^3$ (Northern Rockies) to 0.26 μg/m$^3$ (Ohio

Valley), and underestimated in the southern US, with MB ranging from −0.01 μg/m$^3$ (Southwest) to −0.64 μg/m$^3$ (West).

Concentrations of $SO_2$ are generally overestimated except in the western regions (West, Southwest, and Northern Rockies)

and strong correlations are found throughout all regions. The model underestimates concentrations of $NH_4$ across all regions,

with MB ranging from −0.02 μg/m$^3$ (Northern Rockies) to −0.22 μg/m$^3$ (Ohio Valley). Correlation coefficients for $NH_4$

concentrations are generally strong ($r^2$>0.80) everywhere, except the West and Southwest ($r^2$=0.09 and 0.39, respectively).

Similarly, the model underestimates $SO_4$ concentrations in most regions, with MB ranging from −0.08 μg/m$^3$ (West) to

−0.58 μg/m$^3$ (Southeast), while overestimating in the Northern Rockies (MB=0.07 μg/m$^3$) and Southwest (MB=0.01 μg/m$^3$).

Correlation coefficients for $SO_4$ are also generally strong ($r^2$>0.80), with exceptions in the Southwest and West ($r^2$=0.59 and

0.31, respectively). The EQUATES simulations reproduce the decreasing trend in observed concentrations with strong

 

correlations (r² >0.79) (Figure S6), but overpredict trends of SO₂ and TNO₃ concentrations and underpredicts trends of NH₄ and SO₄ concentrations.

**Figure 4. Scatter plots comparing 2002-2017 annual average concentrations (µg/m³) of sulfur dioxide (a, SO₂), sulfate (b, SO₄), total nitrate (c, TNO₃), and ammonium (d, NH₄) between EQUATES output and CASTNET observations (N=1175). The marker color corresponds to the climate region. Data from CASTNET sites with at least 75% valid data available for each of the 13 years analyzed are included in the analysis shown.**

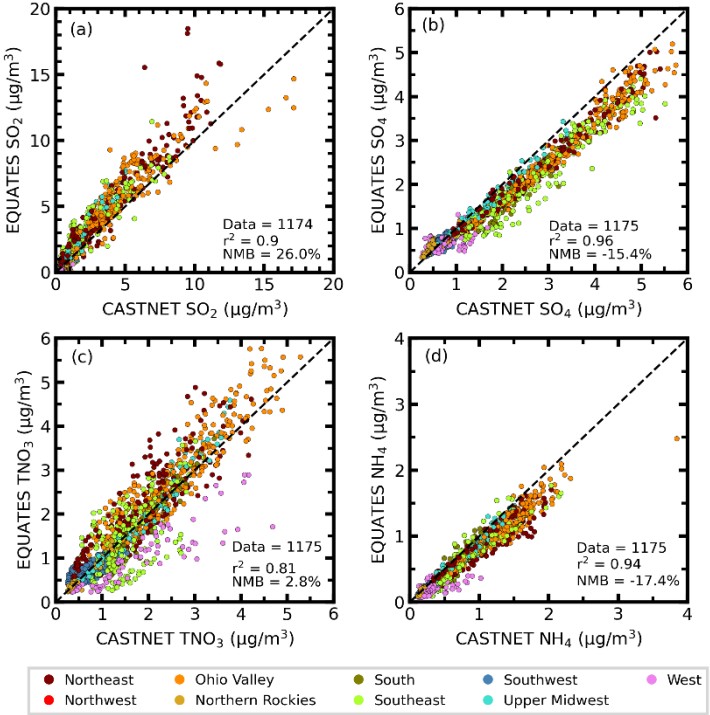

**Table 3. Evaluation of EQUATES modeled average concentrations (µg/m³) of SO₂, SO₄, TNO₃, and NH₄ compared to CASTNET measurements in the CONUS and eight climate regions (no CASTNET data are available in the Northwest climate region).**

| | | SO₄ | | | | SO₂ | | | | NH₄ | | | | TNO₃ | | | |
|---|---|---|---|---|---|---|---|---|---|---|---|---|---|---|---|---|---|
| Climate Region | Sites (N) | Mean Obs/Mod (µg/m³) | MB (µg/m³) | NMB (%) | r² | Mean Obs/Mod (µg/m³) | MB (µg/m³) | NMB (%) | r² | Mean Obs/Mod (µg/m³) | MB (µg/m³) | NMB (%) | r² | Mean Obs/Mod (µg/m³) | MB (µg/m³) | NMB | r² |
| United States | 75 (1175) | 2.15/1.82 | −0.33 | −15.4 | 0.96 | 2.23/2.81 | 0.58 | 26.0 | 0.9 | 0.78/0.64 | −0.14 | −17.4 | 0.94 | 1.72/1.77 | 0.05 | 2.8 | 0.81 |





| | | | | | | | | | | | | | | | | |
|---|---|---|---|---|---|---|---|---|---|---|---|---|---|---|---|---|
| **Northeast** | 15 (237) | 2.40/2.06 | —0.34 | —14.3 | 0.97 | 3.38/4.40 | 1.02 | 30.3 | 0.91 | 0.87/0.72 | —0.15 | —17.8 | 0.91 | 1.77/2.01 | 0.24 | 13.6 | 0.79 |
| **Ohio Valley** | 18 (284) | 3.16/2.66 | —0.50 | —15.9 | 0.93 | 3.81/4.69 | 0.88 | 23.2 | 0.85 | 1.23/1.01 | —0.22 | —17.8 | 0.92 | 2.52/2.79 | 0.26 | 10.5 | 0.86 |
| **Northern Rockies** | 5 (79) | 0.55/0.62 | 0.07 | 12.8 | 0.82 | 0.47/0.44 | —0.03 | —6.95 | 0.37 | 0.22/0.20 | —0.02 | —8.78 | 0.83 | 0.55/0.57 | 0.02 | 4.23 | 0.87 |
| **South** | 6 (94) | 2.37/1.86 | —0.50 | —21.2 | 0.92 | 1.05/1.47 | 0.41 | 39.3 | 0.78 | 0.74/0.63 | —0.11 | —14.8 | 0.85 | 1.69/1.59 | —0.10 | —5.98 | 0.9 |
| **Southeast** | 12 (185) | 2.73/2.16 | —0.58 | —21.1 | 0.93 | 2.10/2.68 | 0.58 | 27.6 | 0.86 | 0.81/0.67 | —0.14 | —17.4 | 0.92 | 1.56/1.48 | —0.07 | —4.76 | 0.4 |
| **Southwest** | 8 (126) | 0.61/0.61 | 0.01 | 13.4 | 0.59 | 0.36/0.34 | —0.02 | —5.61 | 0.76 | 0.22/0.18 | —0.04 | —18.7 | 0.39 | 0.79/0.78 | —0.01 | —1.35 | 0.59 |
| **Upper Midwest** | 5 (78) | 1.77/1.64 | —0.13 | —7.20 | 0.97 | 1.68/2.50 | 0.82 | 49.1 | 0.96 | 0.83/0.75 | —0.08 | —10.1 | 0.95 | 1.98/2.06 | 0.09 | 4.34 | 0.95 |
| **West** | 6 (92) | 0.78/0.70 | —0.08 | —10.9 | 0.31 | 0.40/0.21 | —0.19 | —47.3 | 0.09 | 0.31/0.19 | —0.11 | —37.2 | 0.38 | 1.55/0.91 | —0.64 | —41.6 | 0.79 |

## 3.2 Trend Analysis

### 3.2.1 Total Deposition Trends

Figure 5 shows the spatial distribution and overall trend of modeled total deposition of N (i.e., the sum of oxidized and reduced N) and S between 2002 and 2017. For the trend analysis shown here and throughout, we calculate a linear least-squares regression with significance examined at the 95% confidence level using a Wald Test, with any insignificant trends removed from the analysis. The eastern US has higher total N deposition amounts than the western US in both 2002 and 2017, particularly in parts of the Ohio Valley and Upper Midwest, regions associated with large N emissions from agriculture (Dammers et al., 2019) and energy consumption (Kim et al., 2006). The major N deposition hotspots shift from the eastern to the central US between 2002 and 2017, with highest deposition amounts found in parts of Iowa, North Carolina, and Indiana, a pattern consistent with N inventories compiled by Sabo et al. (2019) indicating an increase in agricultural fertilizer and livestock waste in the Midwest between 2002 and 2012. Increases in agricultural fertilizer have previously been linked to cropland expansion in the Great Plains (Lark et al., 2015; Wright et al., 2017) and increasing demand for domestically sourced biofuels (Donner and Kucharik, 2008).



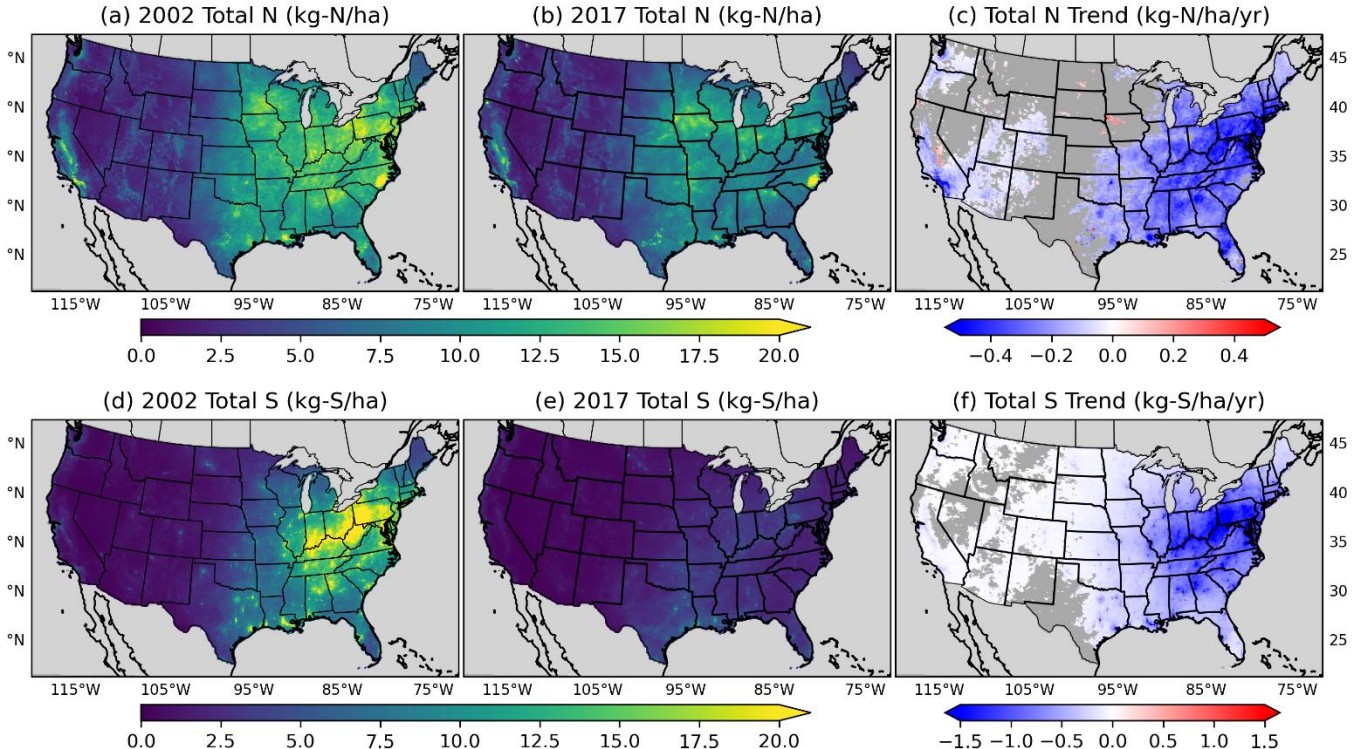

**Figure 5. Spatial distribution of total N (top) and S (bottom) deposition in 2002 (a and d, kg/ha), 2017 (b and e, kg/ha), and the 2002-2017 annual trend (c and f, kg/ha/yr) with significance at the 95% confidence level. Grey areas in panels (c) and (f) indicate where the trend is unavailable or not significant (i.e., p-value of the Wald test is greater than 0.05).**

275 Similar to total N deposition, total S deposition has a noticeable spatial gradient in the east compared to the west, particularly in the beginning of the timeseries, following larger changes in emissions in these regions (Aas et al., 2019; Holland et al., 1999). In 2002, total S deposition across the Southeast, Ohio Valley, and Northeast is broadly greater than 10 kg-S/ha, whereas S deposition for most of the western and central US climate regions is less than 5 kg-S/ha. By 2017, total S deposition continues to be highest in the eastern US, although total S deposition has decreased to less than 7.5 kg/ha on

280 average. As the primary ion in acid deposition, S has been shown to reduce the biodiversity of natural ecosystems (Clark et al., 2019), but is also an essential macro-nutrient required for crop yield and health (Hinckley et al., 2020). Model projections spanning a range of future climate emission scenarios at the end of the century project global S deposition to agricultural soils to decrease by 70-90% compared to 2005-2009 (Feinberg et al., 2021). Consequences of declining S deposition under growing food demand include risk of S deficiencies unless mitigating fertilizer strategies are developed and

285 implemented. The spatial changes in decreasing total N and S deposition tracks changes in $SO_2$ and $NO_x$ emissions since 1970 (Nopmongcol et al., 2019), demonstrating the success of air quality mitigation policies implemented to achieve the NAAQS under the CAAA (United States Congress, 1990) in also reducing nutrient and acid deposition. Concentrations of $SO_2$, a primarily directly emitted compound, indicate greater decreasing trends than the secondary formed $SO_4$. The non-



linear relationship between $SO_2$ and $SO_4$ trends is shown in both the measurements and model output, as well as noted in
other studies using several global CTMs (Aas et al., 2019) and monitoring data (Sickles and Shadwick, 2015), which
attributed the increasing oxidative capacity of the atmosphere more efficiently converting $SO_2$ to $SO_4$ (Sickles and Shadwick,
2015). As $SO_2$ emissions decline from coal burning power plants due to controls implemented to meet the National Ambient
Air Quality Standards (NAAQS) (United States Congress, 1990), additional oxidants are available to oxidize $SO_2$. The
decline in $SO_2$ emissions accompanied by near-constant $NH_3$ emissions results in less acidic cloud droplets (Redington et al.,
2009; Pye et al., 2020), which increases the oxidation rate of $SO_2$ via the ozone pathway (Paulot et al., 2017). Lastly, the
trends in wet deposition of $SO_4$ (Figure S5) are larger than the trends in concentration (Figure S6), as $SO_4$ is more efficiently
scavenged by precipitation.

Figure 6 summarizes average total N and S trends across the climate regions and CONUS based on three different time bins:
2002-2017, 2002-2009, and 2010-2017. While there are significant regional deposition composition changes from 2002-
2017, smaller changes in the overall deposition trends are observed for the CONUS. From 2002 to 2017, the largest average
trend in decreasing total N deposition (─0.19-0.31 kg-N/ha/yr) occurs in the Upper Midwest, Ohio Valley, Northeast, South,
and Southeast. Similarly, the Ohio Valley and Northeast experience the fastest rate of total S deposition decline from 2002 to
2017 (─0.80 kg-S/ha/yr), while the West, Northwest, Northern Rockies, and Southwest indicate near-zero (─0.05 kg-
S/ha/yr) rates of decline. In the Ohio Valley, Northeast, and Southeast, a larger decreasing trend in total N and S deposition
is found between 2002-2009 than 2010-2017, however even the smaller decreasing trend in these regions between 2010-
2017 is larger than  the trend for other regions for all time bins. The high amount of total N and S deposition found in these
regions combined with the slower rate of decline in recent years suggests additional emission controls may be necessary to
continue deposition declines. On the other hand, the Upper Midwest and South present a small increasing trend of ~0.10 kg-
N/ha/yr of total N deposition with large variability from 2010-2017 and a larger declining trend of ─0.34 to ─0.44 kg-
N/ha/yr from 2002-2009. Fertilizer use in the Midwest has only grown modestly (~1.3 %/yr), so increases in total N
deposition have been largely attributed to increasing reduced N deposition caused by warming temperatures and increasing
$NH_3$ remaining in the gas phase due to $NO_x$ and $SO_2$ emission reductions (Warner et al., 2017). As the world's demand for
food grows and $NH_3$ emissions from agriculture increase, models project increasing Nr deposition in many areas across the
globe by 2050 (Paulot et al., 2013) and 2100 (Lamarque et al., 2011).



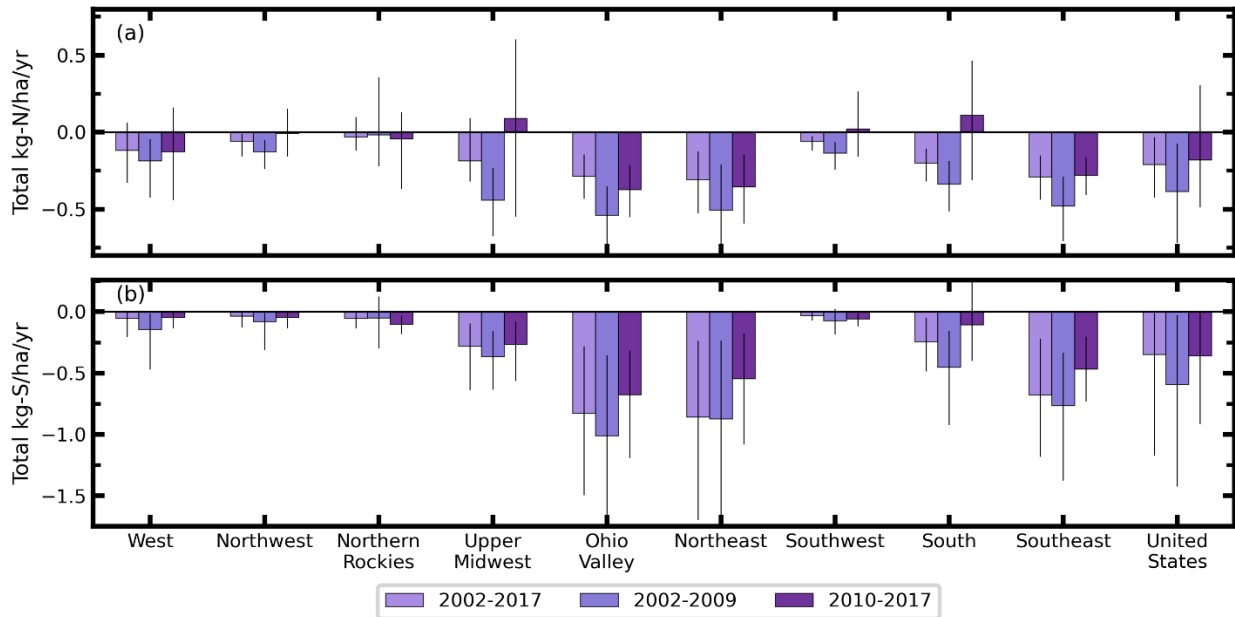

**Figure 6. Comparison of mean total N (a) and total S (b) deposition trends in the nine CONUS climate regions from 2002-2017, 2002-2009, and 2010-2017. Lines on the bars extend to the 5th and 95th percentiles of the trends. Trends only include locations where significant with 95% confidence (i.e., p-value of the Wald's test is less than 0.05, data size is generally much greater than 1000 for all regions except the Northern Rockies where the data size is >250).**

To elucidate the chemical drivers responsible for the varying changes in total N deposition across the US, we examine trends in total oxidized and reduced N in Figure 7. Overall, most regions show larger decreasing trends in oxidized N than increasing trends in reduced N. Regions with insignificant total trends (see grey areas in Figure 5) indicate similar magnitudes of oxidized and reduced N trends. The largest decreasing trends in total oxidized N are found in the eastern US and along the Pacific Coast. The large declines in total oxidized N in the Upper Midwest, Ohio Valley, Northeast, and Southeast showcase both the coordinated application of emission controls in major contributing regions and increasingly lower amounts of $NO_x$ transported from source to receptor areas (Lloret and Valiela, 2016). Unlike trends of total N deposition (Figure 5), statistically significant declining trends for total oxidized N are found throughout most of the CONUS, including the western US. No east-west spatial gradient is observed for trends in total reduced N deposition. Instead, reduced N deposition increases throughout most of the US, particularly in the Upper Midwest and Ohio Valley at an average rate of 0.18 kg-N/ha/yr, but is masked by the larger decreasing oxidized N trends. Declines in total reduced N deposition are located in Los Angeles, CA, Tulsa, OK, and Baton Rouge, LA, suggesting decreases in $NH_3$ emissions near these cities. High $NH_3$ deposition in Baton Rouge (>10 kg/ha) has been attributed to ammonia plant operations in the area (Guo et al., 2018), while dairy farms (Nowak et al., 2012) and vehicles (Cao et al., 2021) are major sources of $NH_3$ in Los Angeles.





**Figure 7. Maps of 2002-2017 trends of total oxidized (a) and reduced N deposition (kg-N/ha/yr) and comparison of average trends across nine CONUS climate regions (c). Lines in panel (c) show the 5th and 95th percentiles of trends for each region. Areas where the trend is significant with 95% confidence are included here (data size>3500 for oxidized and >2000 for reduced in all regions); grey colors on the map indicate where the trend is unavailable or insignificant and not included in panel (c).**

### 3.2.2 Wet and Dry Deposition Trends

Figure 8 shows trends of the wet and dry components of oxidized and reduced N and S deposition. Average dry deposition of oxidized N is found to decline faster than average wet deposition of oxidized N for all climate regions, as oxidized N is more efficiently removed by dry deposition due to high deposition velocities (Zhang et al., 2012). Larger amounts of dry versus wet deposition across the eastern US in 2002 (Figure S7a and d) and more comparable amounts in 2017 (Figure S7b and e) suggest a shift over time in the contributions of oxidized wet and dry deposition, although the portioning is highly dependent upon annual rainfall and changes to climate. The Southeast, Northeast, and Ohio Valley have the steepest simulated average dry and wet oxidized N deposition trends (approximately −0.3 kg-N/ha/yr and −0.1 kg-N/ha/yr, respectively), consistent



with the observed trends (Figure S8). The NTN sites with the highest mean $NO_3$ deposition, particularly those in the
Northeast, experience the largest decreasing trends between 2002-2009 (−2.2 kg-N/ha/yr) compared to 2010-2017 (−1.0 kg-
N/ha/yr), consistent with the trends shown in Figure 6.

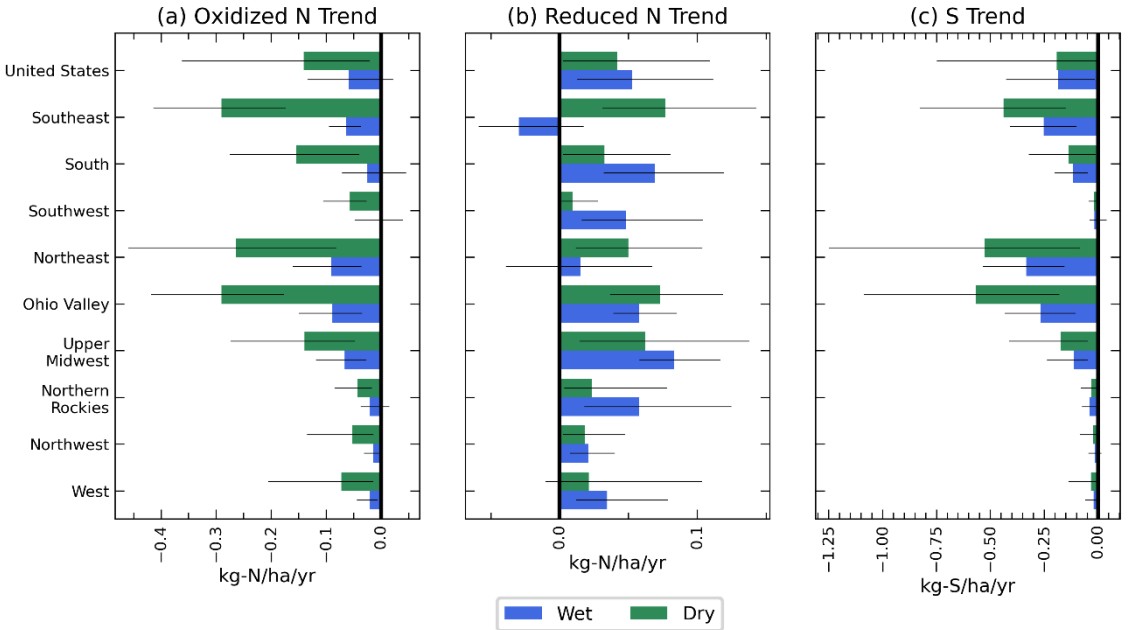

**Figure 8. Annual average trend (2002-2017, kg/ha/yr) of wet (blue) and dry (green) oxidized nitrogen (a), reduced nitrogen (b), and
sulfur (c) deposition throughout the CONUS and nine climate regions. Whiskers extend to the 5th and 95th percentiles of the trends.**
**Cells within each region are included if the annual trend is significant with 95% confidence.**

Trends of reduced N are increasing and overall smaller in magnitude compared to the oxidized N trends (Figure 7b).

Hotspots of wet and dry reduced N deposition have expanded and increased in magnitude across the CONUS (Figure S9)

compared to oxidized N, also observed in the NTN $NH_4$ measurements. The increasing trends of $NH_4$ wet deposition

accompany statistically significant increasing trends in annual precipitation predominantly over the Upper Midwest,
Southwest, and Northern Rockies (Figure S10). Similar increasing trends of $NH_4$ wet deposition between 1989 and 2016

observed in the Midwest and Mid-Atlantic have also been linked to rising annual precipitation amounts (Feng et al., 2021).

The NTN sites with the highest mean $NH_4$ wet deposition experience the largest increasing trend from 2002 to 2017,

especially in the Upper Midwest (0.11 kg-N/ha/yr, Figure S8). Similar statistically significant increases in $NH_4$ wet-

precipitation concentrations have been reported, although increases are largest for the period prior to our model simulations
(1985-1999) (Mchale et al., 2021). The wet component of the reduced N trend is generally larger than the dry, except in the

Ohio Valley and Northeast. The decreasing reduced N deposition in the three urban centers discussed in Section 3.2.1 is

present in the dry deposition trends but not in wet deposition. The Southeast is the only region with a (small) decreasing

trend in wet reduced N deposition and a relatively larger increasing trend in dry reduced N deposition. Parts of the Southeast,



including southwest Florida, experience a decrease in precipitation across the entire period, which may partially explain the
decreasing trend of wet reduced N deposition.

Decreasing trends in wet and dry S deposition are found across the CONUS, although the amount varies by climate region
(Figure 8c). Similar to the oxidized N trends, the Northeast, Southeast, and Ohio Valley experience the largest decreasing
trend of approximately ─0.5 kg-S/ha/yr for dry deposition and ─0.3 kg-S/ha/yr for wet deposition, consistent with the NTN
$SO_4$ observations (Figure S8). Sites with the largest mean $SO_4$ wet deposition in the Ohio Valley and Northeast also
experience the largest decreasing trends in wet $SO_4$ deposition (─1.9 kg-S/ha/yr, Figure S8), consistent with monthly mean
precipitation-weighted concentrations from NTN sites from 1985 to 2017 (Mchale et al., 2021). From 2002 to 2017 there is a
shift from dry deposition to wet deposition as the dominant deposition pathway for S in the eastern US (Figure S11). Large
decreasing trends in dry S deposition are clearly seen over large urban centers in the western US, while the eastern US
experiences a significant declining trend everywhere in response to decreases in regional $SO_2$ emissions (Likens et al., 2001).

**3.3 Total Nitrogen and Sulfur Deposition Budgets**

The average N and S deposition budgets across all CONUS climate regions generally decrease from 2002 to 2017 (Figure 9).
The average total N budget for the CONUS decreases from 7.8 kg-N/ha to 6.3 kg-N/ha between 2002 and 2017. The Upper
Midwest, Ohio Valley, Northeast, and Southeast have the largest average total N, decreasing from 12-14 kg-N/ha in 2002 to
8-10 kg-N/ha in 2017. In contrast, the Northwest, West, and Southwest have the smallest average total N which decreases
only slightly from 3.1-3.9 kg-N/ha to 2.8-3.5 kg-N/ha over the same time period. The decrease in total N deposition is
largely due to reductions in oxidized N, particularly from dry deposition, and declines in $NO_x$ emissions (Nopmongcol et al.,
2019). The average wet oxidized N deposition for the CONUS decreases from 1.6 kg-N/ha in 2002 to 1.1 kg-N/ha in 2017,
while the dry component decreases from 3.7 kg-N/ha to 1.7 kg-N/ha. Dry deposition constitutes most of the total N budget
(slightly decreasing from 50-82% in 2002 to 42-76% in 2017 across the climate regions), with the largest average
contributions found in the western US. Spatial differences in the dry N deposition flux, inferred using Ozone Monitoring
Instrument (OMI) satellite observations of $NO_2$ and ground measurements from 2005 to 2014, indicate an increasing flux
across the western US and decreasing flux over the eastern US due to reductions in $NO_x$ and $NH_3$ emissions (Jia et al., 2016).
Unlike oxidized N, the wet reduced N deposition is larger on average than the dry deposition for all regions except the West
and Northwest, regions that typically experience less rainfall than the other regions. Similar to Du et al. (2014), we find an
increasing contribution of wet reduced N deposition to the total across the CONUS from 19% to 30% across the time period
(Figure S12).

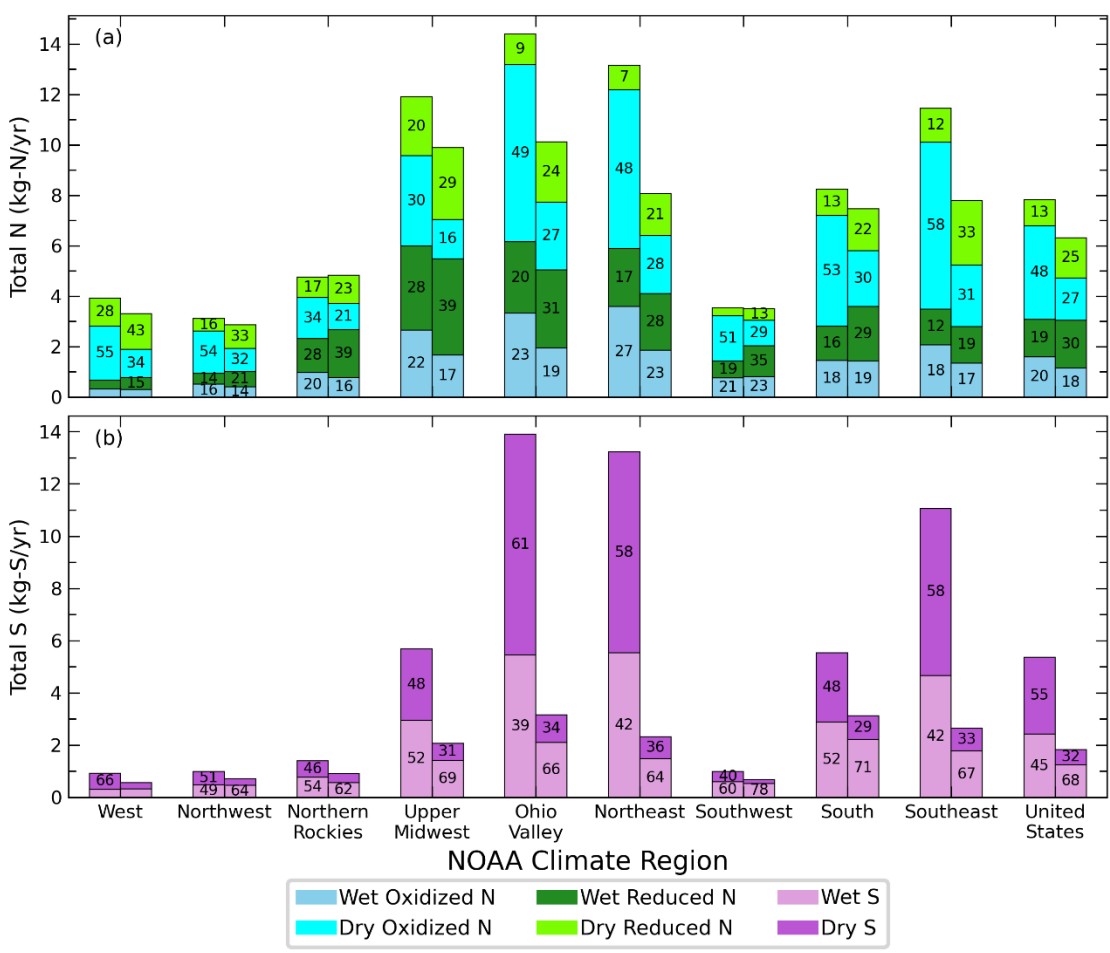

**Figure 9. Average total (wet+dry) N (a) and S (b) deposition in 2002 (left bars) and 2017 (right bars) throughout the CONUS and nine climate regions. Colors denote the wet and dry component of each species. The numbers in each bar denote the percentage contribution to the annual total.**

The relative proportion of reduced N to the total budget has shifted over the 2002 to 2017 time period (Figure 10). On average across the CONUS, oxidized N deposition dominates the total budget until 2016 when reduced N comprises >50% of the total, consistent with previous studies (Du et al., 2014; Kanakidou et al., 2016; Ackerman et al., 2019; Li et al., 2016; Nopmongcol et al., 2019) and attributable to the absence of $NH_3$ emission controls from agricultural sources combined with the success of $NO_x$ regulations. The reductions in $NO_x$ and $SO_x$ emissions due to policy controls result in less $NH_x$ partitioning to the aerosol phase and more to the gas phase, favoring dry deposition closer to source regions. Despite spatial variations in agricultural $NH_3$ sources and lack of an emission trend during this study period over some locations, all climate regions indicate an increasing amount of reduced N to the total budget, agreeing with results from Nopmongcol et al. (2019).





In order to protect critical ecosystems within US national parks from future N deposition, Ellis et al. (2013) project at least a 50% reduction in anthropogenic $NH_3$ emissions is required relative to representative concentration pathway (RCP) projected emission scenario levels in 2050. The fraction of reduced N to the total is largest in regions with larger $NH_3$ emissions (Dammers et al., 2019; Behera et al., 2013), such as the Upper Midwest and Northern Rockies, throughout the time period studied here. These regions exceed 50% of reduced N as a fraction of the total budget as early as 2006 for the Upper

Midwest and 2011 for the Northern Rockies. Approximately 80% of the total global $NH_3$ emissions in 2005 are attributable to agriculture, including animal feedlot operations, and are anticipated to increase in the US and globally (Behera et al., 2013). The increasing importance of reduced N is likely to impact future N deposition budgets and the competitive nature among plants with varying affinities for the different forms of nitrogen (Choudhary et al., 2016; Kahmen et al., 2006).

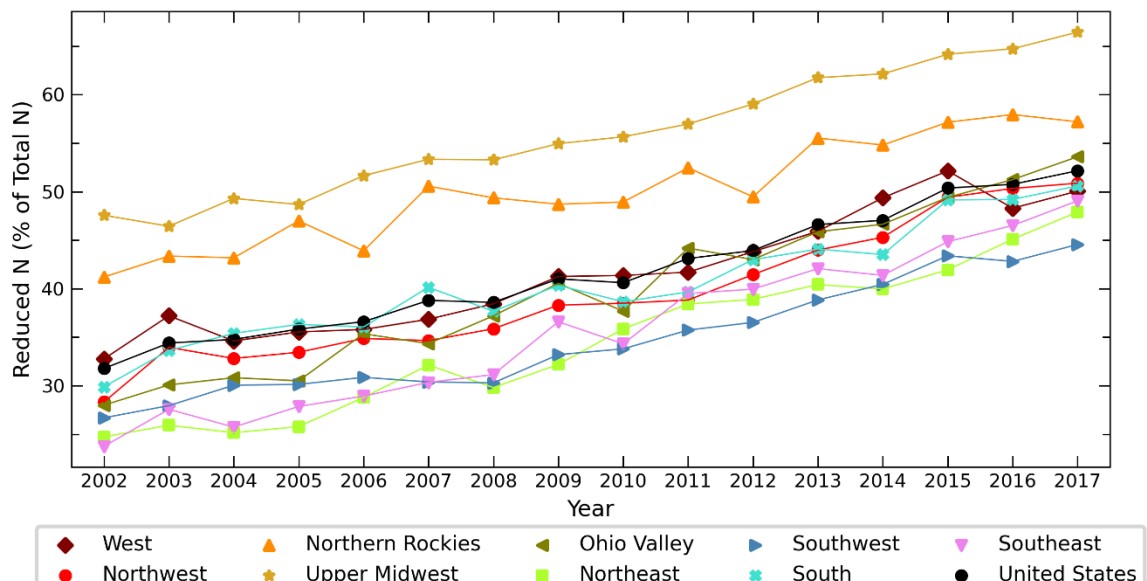

**Figure 10. Changes in the percent of total N deposited as reduced nitrogen from 2002 to 2017 throughout the NOAA climate regions and CONUS.**

The total S budget also decreases across the CONUS on average, falling from 5.3 kg-S/ha in 2002 to 1.8 kg-S/ha in 2017. Decreasing $SO_2$ emissions have been previously shown to be responsible for decreasing S deposition (Nopmongcol et al.,

2019). Similar to the total N budget, the largest average amounts of total S are located in the Ohio Valley, Northeast, and Southeast, decreasing from 11-14 kg-S/ha in 2002 to 8-10 kg-S/ha in 2017. In the West, Ohio Valley, Northeast, and Southeast, dry deposition dominates the budget 2002, while transitioning wet deposition dominated by 2017. Wet deposition comprises a greater portion of the total S budget in 2017 across all climate regions. The primary driver behind the shift in the





total S budget may be emission reductions of $SO_2$ implemented to meet the NAAQS (leading to decreasing ambient
concentrations of $SO_2$ and $SO_4$) that more efficiently decrease the dry component than the wet (Sickles and Shadwick, 2015).

**4 Conclusions**

In this study, we examine CMAQ model simulations from the EPA's Air QUAlity TimE Series (EQUATES) project to
investigate spatial and temporal trends of nitrogen (N) and sulfur (S) deposition over a period with substantial emission
reductions implemented to meet requirements set by the Clean Air Act Amendments (CAAA). We assess changes in
modeled dry deposition and precipitation- and bias- adjusted wet deposition across nine climatologically consistent regions
within the US. EQUATES CMAQ simulations included important science updates to previous modeling studies, such as
revised model for bidirectional air-surface exchange of $NH_3$ and a mechanistic representation of organic N (Pye et al., 2015),
necessary to improve modeling of Nr budgets. The measurement-model fusion technique for modeled wet deposition
estimates reduce the model bias and improve correlations compared to the unadjusted model values, and can thus improve
the accuracy of critical loads assessments based on model data. Comparisons to wet deposition observations from the NADP
network indicate the model generally underestimates wet deposition of $NH_4$, $NO_3$, and $SO_4$ across the CONUS. Agreement is
poor in regions with complex terrain like the Northern Rockies and West, potentially due to errors constraining emissions
and/or reproducing precipitation at 12km grid spacing, as sub-grid variability in precipitation estimates can differ from sub-
grid variability in concentrations across the country. Similarly, the model underestimates ambient concentrations of $NH_4$ and
$SO_4$ compared to CASTNET measurements across the CONUS, while $SO_2$ and $TNO_3$ concentrations are overestimated.
Additional measurements in regions with high bias, such as the West, Northwest, and Southwest, as well as model
improvements to precipitation estimates, particularly in areas of complex terrain, could greatly improve agreement with the
NADP network observations.

The modeled total N and S deposition amounts and trends are larger across the eastern than the western US, particularly in
the Northeast, Southeast, Ohio Valley, and Upper Midwest climate regions. While the average CONUS trend indicates a
decrease of −0.21 kg-N/ha/yr of total N deposition from 2002-2017, significant increasing trends are found in parts of the
Upper Midwest, Northern Rockies, and West. Such increases in total N deposition are dominated by increases in reduced N
that are detected across all climate regions, but are masked by the larger declining trends of oxidized N deposition as a result
of policies targeting $NO_x$ emissions enacted under the CAAA. The increasing trend in reduced N from 2002 to 2017 is a
result of growing agricultural emissions combined with $NO_x$/$SO_x$ emission reductions and higher precipitation amounts over
time concentrated in the Upper Midwest, Southwest, and Northern Rockies. Despite the overall average declining N trend
from 2002 to 2017, the simulations indicate nearly twice as large average decreasing trend of total N and S at the beginning
of the timeseries (2002 to 2009) than the end of the period (2010 to 2017). Total N trends in the Upper Midwest and South
reveal a slight increasing trend (0.15 kg-N/ha/yr) from 2010 to 2017, contrary to the larger declining trends found between

2002-2009. The modeled slowing and slight increasing trend for total N across the climate regions in the EQUATES simulations hint at future changes to the N budget that may impact plant biodiversity and productivity (Kahmen et al., 2006).

The average total N deposition budget over the CONUS has decreased from 7.8 kg-N/ha in 2002 to 6.3 kg-N/ha in 2017, largely due to decreases in oxidized N deposition as a result of $NO_x$ emission controls implemented under the CAAA. The major contributor to the N budget is dry deposition, although the contribution across the climate regions has declined from 50-82% in 2002 to 42-76% in 2017, in part due to changes in precipitation amounts. The regions with the largest average total N deposition, including the Ohio Valley, Northeast, Upper Midwest, and South, also experience the largest reduction in total N deposition. Similar to the total N budget, the average total S budget over the CONUS has declined from 5.3 kg-S/ha

in 2002 to 1.8 kg-S/ha in 2017. The largest amount of total S occurs in the eastern US, where total deposition is more impacted by dry deposition. By the end of the timeseries, wet deposition dominates the S budget for all climate regions, comprising 68% of the total budget in the CONUS on average.

Our analysis, in addition to the analyses of many others (Zhang et al., 2018; Li et al., 2016; Du et al., 2014), highlights the

increasing contribution of reduced compounds to the N budget in all climate regions, particularly the Northern Rockies and Upper Midwest. Both the model and observations indicate statistically significant increasing trends in $NH_4$ deposition across the CONUS due to changes in precursor emissions. Reductions in $NO_x$ and $SO_2$ emissions, and therefore their oxidation products, due to policy controls result in less $NH_x$ partitioning to the aerosol phase and more to the gas phase, favoring dry deposition of $NH_3$ closer to source regions. Additionally, growth of $NH_3$ emissions from agriculture (Behera et al., 2013)

sources allows for additional $NH_4$ aerosol formation in the atmosphere given sufficient $NO_x$ and $SO_x$. This study further confirms the growth of reduced N in recent years across all climate regions, and additionally suggests an attenuation of the declining oxidized N deposition trend. To regulate $NH_3$ emissions to protect human health and lower environmental risk, especially in the agricultural sector, it is imperative to collect $NH_3$ data in areas exceeding the NAAQS for particulate matter and its gaseous precursors. In addition, a more robust characterization of dry deposition in the US, including low-cost dry

deposition and ambient measurement methods, is needed, particularly in agricultural areas and regions with transitions from urban to rural environments, as well as species not routinely measured, such as organic compounds.

**Data Availability**

Data is available for download at https://www.epa.gov/cmaq/equates.



**Author Contributions**

KF and GP were the EQUATES project leads and responsible for project administration, methodology, validation, formal analysis, data curation, and visualization. RG conducted the meteorological runs and WA and CH performed the model simulations. RG, CH, WA, and JB provided methodology, formal analysis, and validation. SB performed the data analysis and created all figures and tables. SB wrote the paper with comments from all authors.

**Disclaimer**

The views expressed in this paper are those of the authors and do not necessarily represent the views or policies of the US Environmental Protection Agency.

**Acknowledgements**

The authors would like to thank Chris Allen, Michael Aldridge, Megan Beardsley, James Beidler, David Choi, Alison Eyth, Caroline Farkas, Janice Godfrey, Barron Henderson, Shannon Koplitz, Rich Mason, Rohit Mathur, Chris Misenis, Norm
Possiel, Havala Pye, Lara Reynolds, Matthew Roark, Sarah Roberts, Donna Schwede, Karl Seltzer, Darrell Sonntag, Kevin Talgo, Claudia Toro, and Jeff Vukovich for their contribution to the development of emissions and meteorology inputs for the EQUATES simulations used in this study. This research was supported in part by an appointment to the Research Participation Program at the US EPA, Office of Research and Development (ORD), administered by the Oak Ridge Institute for Science and Education (ORISE) through an interagency agreement between the US Department of Energy and the US
EPA.

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
