# Peer review of "Long-term Regional Trends of Nitrogen and Sulfur Deposition in the United States from 2002 to 2017"

_Atmospheric Chemistry and Physics, 2022_

## Referee Comment (RC1)

**Review of Long-term Regional Trends of Nitrogen and Sulfur Deposition in the United States from 2002-2017**

**General Comments**

In this manuscript, Benish and co-authors provide updated model estimates of wet, dry, and total N and S deposition across the continental United States. As S deposition declines and N deposition transitions from being dominated by oxidized forms to reduced forms, this type of measurement-model fusion is extremely important. With some revision, this manuscript will make an important contribution to the atmospheric deposition literature, and the model estimates will be very useful to many scientific and stakeholder communities. I will be excited to see this published.

I have three broad concerns about the manuscript, in addition to more specific questions and comments listed below. First, there is almost no discussion of urban emissions and deposition in this manuscript, although the final sentence does include a note that more measurements are needed in regions with transitions from urban to rural environments. There has been a lot of recent literature on urban hotspots of N deposition, and this is a growing area of focus within NADP and TDEP. Given the completeness criteria described in the methods, none of the urban NADP sites would have been included in this study, although it would be extremely interesting. It is not clear to me how the EQUATES model handles urban emissions and deposition, but most urban centers are not visibly apparent in the Figure 5 maps, even though we know that cities have high deposition values. Even if urban areas are not explicitly included in the analyses, I think it is important to mention urban deposition more frequently throughout the manuscript, and to explain how urban areas may contribute to model uncertainty.

Second, I'm left wondering why the dry deposition does not undergo the same model-measurement fusion that is used for the wet deposition. I understand that it would be impossible to do this with actual deposition estimates, but it seems like there are some clear biases in the modeled concentration values (shown in Figure 4) that could be corrected with a similar measurement-model fusion process. If there is a good reason that this works well for wet deposition but not for dry deposition, this should be stated explicitly in the text.

Finally, I am wondering how the focus on annual values plays into some of the measurement-model mismatch. While annual values are used very frequently, they hide the extreme seasonality of atmospheric deposition. How well is this seasonality captured in the EQUATES model? I understand that a full exploration of seasonal patterns would be another manuscript, but I am curious if model biases in both precipitation estimates and concentration estimates are season-dependent. Some discussion of the focus on annual values would be helpful.

**Specific Comments**
*Abstract* – Overall, the abstract is quite long and I found it challenging to follow. I think it could benefit from a clearer structure.
Line 28: It is unclear how regional $NO_x/SO_x$ emission reductions contribute to widespread increases in reduced N deposition. I understand that the reduction in $NO_x$ deposition increases

the proportion of N deposited in reduced form, but it seems like this is stating that is contributes to the absolute increase.

*Introduction*
Line 45: This seems like a limited definition of dry deposition, because dry deposition could be deposited on surfaces other than leaves, like soil or water.
Line 57: In my opinion, urban areas and intense agricultural areas are also essential locations that have limited measurements.
Lines 75-79: I am confused by the relationship between this project and TDEP, and I would like to see more comparison to TDEP products throughout the manuscript. Is this effort part of TDEP, or will the results here be incorporated into the TDEP products? TDEP products are used extensively by the NADP community, so clarification here would be very helpful.

*Methods and Materials* – Throughout the methods and materials, it would be helpful to be extremely clear about the timescale used. It seems like most calculations were done on an annual basis, but this was sometimes confusing.
Line 121: Was this calculation correcting for chemical transformations performed on an annual basis? Or on a weekly basis?
Equation 1: The precipitation correction is done on an annual basis, which seems like it could be problematic. Because N deposition has such a strong seasonal cycle, it matters when the precipitation is either over- or underestimated. If the modeled precipitation is too low mostly during the winter when N concentrations are low, an annual correction could then overestimate N deposition. For more discussion of this problem (and how it introduces error into the NADP annual estimates), see (Schichtel et al., 2019). It would be helpful to see some discussion about the decision to focus on annual values and the issues that this may introduce into the calculations.
Line 148: How sensitive is this method to the 300-km radius? How was this radius chosen?

*Results and Discussion*
Figure 3: Is the precipitation here from PRISM or from NTN rain gauges? I don't think the NTN precipitation depth measurements were mentioned in the sampling method section, so this might be confusing to people who are unaware that NTN measures precipitation depth.
Lines 206-224: I found it confusing that this paragraph mixes general results on spatial variability in total deposition with model performance.
Lines 264-267: How are you defining hotspots here, and how were they identified? There are many urban areas that are known hotspots of N deposition (e.g., Denver-Boulder metro area), but these do not appear on the map in Figure 5 (but perhaps this is because of the spatial or color scale?).
Line 312: Explain the connection between warming temperatures and increasing reduced N deposition more fully. Also, what about on-road emissions of ammonia? These are an increasingly important source of $NH_3$ emissions connected to $NO_x$ emission control mechanisms (Fenn et al., 2018).
Figure 5: It might also be helpful to distinguish between areas with unavailable and not significant trends in Figure 5, because these have very different meanings. In this trend analysis,

is it possible to have a significant trend with a slope of zero? Figure 5f appears very white – are these places with very small significant trends, or is the slope actually zero?

Figure 6: I am struggling to interpret Figure 6, given the fact that areas without a significant trend are removed. Judging by Figure 5, it seems like this removes the vast majority of many regions. If there is a small decreasing trend in a corner of a region that generally has had stable N deposition, it seems misleading to represent that as a decreasing trend for the whole region. I'm also confused by what 'data size' refers to in the caption. Is each data point a pixel on Figure 5?

Line 356: Again, I'm curious how you are defining the term 'hotspot.'

**Technical Corrections**

Line 54: Rephrase so it is clear that the NADP, rather than wet deposition, is the subject of the verb "collecting."

Line 174: Define NMB in the text as well as in the figure captions.

Figure 2: It would be helpful to make the dashed and dotted lines more visibly different.

**References**

Fenn, M. E., Bytnerowicz, A., Schilling, S. L., Vallano, D. M., Zavaleta, E. S., Weiss, S. B., et al. (2018). On-road emissions of ammonia: An underappreciated source of atmospheric nitrogen deposition. *Science of The Total Environment*, *625*, 909–919. https://doi.org/10.1016/j.scitotenv.2017.12.313

Schichtel, B. A., Gebhart, K. A., Morris, K. H., Cheatham, J. R., Vimont, J., Larson, R. S., & Beachley, G. (2019). Long-term trends of wet inorganic nitrogen deposition in Rocky Mountain National Park: Influence of missing data imputation methods and associated uncertainty. *Science of The Total Environment*, *687*, 817–826. https://doi.org/10.1016/j.scitotenv.2019.06.104

---

## Referee Comment (RC2)

In this study, the authors investigated the long-term trend of deposition of N and S using state-of-the-art regional CMAQ model. Model evaluations for the deposition as well as concentration for specific air pollutants are reasonable. The conclusions are not surprising that the depositions in the US are declining from 2002 to 2107, with contributions of reduced nitrogen increasing and oxidized nitrogen decreasing, which are consistent with several previous studies. In general, this study was well designed and fit into the journal. The authors need to make efforts to improve the reading flow for the manuscript, as well as to improve their quality of figures and tables.

Change the hyphen "—" to minus "−" through the whole manuscript.

**Abstract:**
Line 11-13 "few assess dry deposition, incorporate a measurement-model fusion approach to improve wet deposition estimates, or focus on changes within specific US climate regions." This was exactly what was covered in my two previous studies (Zhang et al., 2018, 2019) which was cited by the authors as well. I suggest the authors refine their motivation or novelty for this study. Also, read the latest paper by Tan et al. (2020) and distinguish the novelty between this study with previous one.

Reference:
Tan, J., Fu, J. S. and Seinfeld, J. H.: Ammonia emission abatement does not fully control reduced forms of nitrogen deposition, Proc. Natl. Acad. Sci. U. S. A., 117(18), 9771–9775, doi:10.1073/pnas.1920068117, 2020.

Line 16—17: Reading from section 2.1, the authors state that the STAGE option was performing similar results as M3dry. So I did not see the point/novelty for the authors to add this statement in the abstract. Also abbreviation for "STAGE" is not necessary since it was not referred again in the abstract.

Line 22: Explain "$TNO_3$"

Line 22-23: Is this sentence used to explain the model evaluation of wet deposition, or concentration?

Line 27: Will the "increased precipitation" increase both the reduced and oxidized N deposition as well?

Line 29-30: This is an interesting finding. Can the author provide explanations why this happens?

Line 30: change to "The average annual total N"?

**Introduction:**

Line 69: define "$TNO_3$" and "$NH_X$"

Line 76: "TDEP" to "TDep"

Line 97: Please reorganize this sentence. The Hemisphere CMAQ was used to provide BCs for the 12 km CMAQ only, but not used for the data analysis in this study.

Line 101: "STAGE" was already defined.

**Methods**
Section 2.2: Why the authors explain why the criteria for NTN and CASTNET differ with each, "at least 60% annual coverage" for NTN, and "75% annual coverage" for CASTNET?

**Results and Discussions**
Line 161: define "ECODEP"

Line 167: Please provide figure/table for this statement "although the EQUATES precipitation is still biased low on average relative to PRISM."

Line 219-224: Please show a plot/table for this conclusion. Also, discuss the $NH_4$ first and then $NO_3$ and $SO_4$, following the flow of earlier discussions in the same paragraph.

Line 206: "The 16-year total $NH_4$": is this the 16 year total or 16 year annual average? The same applies to the $NO_3$ and $SO_4$.

Line 234-line 245: I suggest the authors to follow the order of "$NH_4$, $TNO_3$, $SO2$, and $SO_4$" when discussing the model performances of the concentration.

Line 287: define NAAQS here instead of in line 293.

Line 302: "—0.19-0.31 kg-N/ha/yr)": Is the 0.31 positive or negative trend?

**Figures & Tables**
Figure 1: "Site locations of the 200" Reading from the figure, I believe the authors mean "the 263" NADP locations instead of 200 since they have "black-bordered white circles" vs. "black circles"?

Figure 2: "black circles"—I think the authors meant the 200 "black-bordered white circles"?
In the legend, there are lines associated with the rectangle and diamond, while there are none in the Taylor plot.

Table 2: Put Table 2 in Landscape orientation, which will make the table look much better. The same as Table 3.

Figure 9: change to "throughout the nine climate regions and CONUS". Also change "United States" in the bottom bar to "CONUS";

Fig. 9(b) Missing the percentage contribution for the year 2002 in "West", "Southwest", and also for the year 2017 in "West"
Fig. 9(b) Missing the percentage contribution for the year 2017 in "West", "Northwest", "Northern Rockies", and "Southwest";

Figure 10: Be consistent for the usage of "CONUS" and "United States"

---

## Author Comment (AC1)

The response to Referee comments is in blue.

In this manuscript, Benish and co-authors provide updated model estimates of wet, dry, and total N and S deposition across the continental United States. As S deposition declines and N deposition transitions from being dominated by oxidized forms to reduced forms, this type of measurement-model fusion is extremely important. With some revision, this manuscript will make an important contribution to the atmospheric deposition literature, and the model estimates will be very useful to many scientific and stakeholder communities. I will be excited to see this published.

We thank Dr. Heindel for the helpful comments that have improved the manuscript. Specific responses to each comment are below.

I have three broad concerns about the manuscript, in addition to more specific questions and comments listed below. First, there is almost no discussion of urban emissions and deposition in this manuscript, although the final sentence does include a note that more measurements are needed in regions with transitions from urban to rural environments. There has been a lot of recent literature on urban hotspots of N deposition, and this is a growing area of focus within NADP and TDEP. Given the completeness criteria described in the methods, none of the urban NADP sites would have been included in this study, although it would be extremely interesting. It is not clear to me how the EQUATES model handles urban emissions and deposition, but most urban centers are not visibly apparent in the Figure 5 maps, even though we know that cities have high deposition values. Even if urban areas are not explicitly included in the analyses, I think it is important to mention urban deposition more frequently throughout the manuscript, and to explain how urban areas may contribute to model uncertainty.

We agree expanding the discussion of urban deposition in EQUATES would better highlight the data need for deposition measurements in and near cities. There have been several recent publications documenting N deposition in urban areas at approximately twice that of their surrounding landscape (Decina et al., 2019). The observed urban deposition increase is composed primarily of reduced N. In CMAQ we do have higher urban emissions particularly for NOx and do estimate mobile NH3 emissions, but these may be underestimated (Sun et al., 2017). The urban enhancement is not readily visible on the maps due to the agricultural deposition enhancement which can be an order of magnitude higher than the surrounding non-agricultural areas (Shen et al., 2016; Walker et al., 2014) and due to model resolution, which is larger than many urban centers.

We make the following changes to address this comment, including clarifying some of the methodology and expanding the analysis:

1. Adding relevant literature to the second paragraph in the introduction. The revised text now reads:
   Despite providing critical deposition information, the limited number of NADP and CASTNET sites in essential locations, such as areas with complex terrain, near urban centers, at high elevation, or in forest ecosystems, restrict a thorough understanding on the amount and consequences of deposition. For instance, strong concentration gradients in N deposition have been documented over urban areas such as Boston, MA (Rao et al., 2013) and near Baltimore, MD (Bettez et al., 2013), and along coastlines like the Chesapeake Bay (Loughner et al., 2016).
2. Clarifying methodology to reflect the use of urban NADP sites. Urban sites are included in the measurement model fusion technique for each year, but many of these sites are online for short periods of time (including several sites in Colorado: CO06, CO84 ,CO85, CO86, CO87), so not evaluated in the multi-year trend analysis. Urban sites that are evaluated in Section 3.1 include KY19, MD99, NC41, and NJ99 and we extend Table S1 to include the NADP reported site class (urban, suburban, rural, or isolated). Text in the methods is expanded to reflect this:
   After adjusting simulated wet deposition by precipitation, an additional bias-adjustment (EQUATES$_{bias-adj}$) is applied using all NTN observations that meet annual data completeness, which varies year to year and includes sites of all classifications.
3. Add revised maps of 2017 total oxidized and reduced N deposition for easier viewing of urban sources to supplemental information (now Fig S8, see below). Added text in Section 3.2.1 reads:

Urban regions in the central and eastern US indicate a substantial amount of N deposition compared to nearby rural areas (Figure S7), consistent with previous findings that bulk N deposition in urban areas is twice as much as rural and remote sites (Decina et al., 2019).

[Figure]

4.  Adding text in conclusions regarding modeling uncertainty in urban areas:

    However, since N deposition over urban areas across the CONUS is likely already underestimated (Rao et al., 2013), with increasing urbanization only expected to further increase N deposition amounts (Joyce et al., 2020), addressing modeling uncertainty in emissions and chemistry at relevant spatial and temporal resolutions is imperative.

Second, I'm left wondering why the dry deposition does not undergo the same model measurement fusion that is used for the wet deposition. I understand that it would be impossible to do this with actual deposition estimates, but it seems like there are some clear biases in the modeled concentration values (shown in Figure 4) that could be corrected with a similar measurement-model fusion process. If there is a good reason that this works well for wet deposition but not for dry deposition, this should be stated explicitly in the text.

Measurement-model fusion methods have been widely used to improve modeled concentration output, but have been more limited for wet deposition. The NADP Total Deposition (TDep) science committee applies a fusion technique to improve dry deposition estimates with measurements from CASTNET, AMoN, and SEARCH, but presently only adjusts wet deposition by precipitation (Schwede and Lear, 2014). Zhang et al. (2019) addressed this science need by constructing an approach to bias-correct CMAQ model simulations based on observed precipitation and wet deposition observations. The TDep dry deposition fusion products are only available for an older timeseries (ECODEP) and is undergoing methodology improvements (see https://nadp.slh.wisc.edu/committees/tdep/). Efforts incorporating a wet deposition bias correction to TDEP is ongoing, but unfortunately not available yet.

To address this comment, we add the following text in the introduction paragraph about deposition measurements:

    Dry deposition modeling is still uncertain, particularly for land use and dry deposition schemes in models and emissions data.

In addition, we add the following text in the introduction paragraph about modeling atmospheric deposition:

    As such, the NADP Total Deposition Science Committee (TDEP, see www.nadp.slh.wisc.edu/committees/tdep) advances methods to improve estimates of atmospheric deposition from the Community Multiscale Air Quality model (https://www.epa.gov/cmaq) (see Schwede and Lear (2014)). TDEP products only employ a fusion approach to dry deposition and currently use an older version of CMAQ, although efforts to update the model version and incorporate wet deposition fusion described herein are ongoing.

Additionally, we revise the first sentence in Section 2.3 to more clearly express reasoning behind the inability to correct the dry deposition estimates:

The modeled wet deposition fields are adjusted to account for input biases and uncertainty in the chemical and physical processes governing deposition, but not applied to dry deposition due to limited dry deposition measurements.

Finally, I am wondering how the focus on annual values plays into some of the measurement-model mismatch. While annual values are used very frequently, they hide the extreme seasonality of atmospheric deposition. How well is this seasonality captured in the EQUATES model? I understand that a full exploration of seasonal patterns would be another manuscript, but I am curious if model biases in both precipitation estimates and concentration estimates are season-dependent. Some discussion of the focus on annual values would be helpful.

We agree that annual values hide the extreme seasonality of atmospheric deposition. Zhang et al. (2019) previously tested the seasonality of the measurement-model fusion, finding:

1. The model overestimates precipitation, except during fall, with correlations ($R^2$) lowest in summer (0.47) and highest in winter (0.65). The correlation for the annual accumulated precipitation is higher (0.67).
2. $NO_3$ wet deposition is overestimated by the model in winter (8.6%) and fall (12.4%) and underestimated in spring (-8.2%) and summer (-6.7%). The normalized mean bias (NMB) of the annual accumulation values is -1.2%. The correlation for seasonal $NO_3$ wet deposition between the model and measurements is highest in spring (0.73) and lowest in summer (0.54), with higher annual $R^2$ of 0.76.
3. The model underestimates $NH_4$ wet deposition for all seasons except summer (NMB=12.6%). The annual NMB is -11%. The correlation for seasonal accumulated $NH_4$ wet deposition is below 0.5, except for spring (0.55) and the annual accumulated values (0.60).
4. $SO_4$ wet deposition is only overestimated in summer (2.0%). The annual NMB is -4.5%. The correlation is comparable for all seasons (0.69-0.71), except for slightly lower values in winter (0.57).

The table below summarizes the uncorrected seasonal wet deposition and precipitation summary statistics from EQUATES (2002-2017). This table will be added to the supplementary information (Table S3).

| | | Winter | Spring | Summer | Fall | Annual, no adjustment (Table 1) |
|---|---|---|---|---|---|---|
| $NO_3$ | $r^2$ | 0.63 | 0.74 | 0.60 | 0.69 | 0.77 |
| | MB (kg/ha) | 0.01 | 0.09 | 0.04 | 0.23 | 0.55 |
| | NMB (%) | 1.37 | 5.44 | 2.22 | 20.90 | 9.64 |
| $NH_4$ | $r^2$ | 0.50 | 0.63 | 0.49 | 0.55 | 0.61 |
| | MB (kg/ha) | -0.10 | -0.23 | -0.03 | -0.10 | -0.49 |
| | NMB (%) | -39.7 | -34.8 | -3.70 | -25.0 | -19.9 |
| $SO_4$ | $r^2$ | 0.65 | 0.71 | 0.72 | 0.67 | 0.78 |
| | MB (kg/ha) | -0.17 | -0.26 | -0.30 | -0.17 | -0.92 |
| | NMB (%) | -18.9 | -14.3 | -14.5 | -12.9 | -12.2 |
| Precipitation | $r^2$ | 0.77 | 0.70 | 0.61 | 0.69 | 0.72 |
| | MB (kg/ha) | 3.63 | -2.66 | -12.9 | -23.2 | -2.32 |
| | NMB (%) | 2.28 | -1.30 | -5.88 | -11.7 | -2.40% |

From Zhang et al. (2019) and the table above, we find the seasonal correlation values are fairly low in fall and winter in multiple regions (see Figure S1 in Zhang et al. (2019)), therefore a seasonal precipitation adjustment will be less effective. The annual deposition and precipitation values are used in the measurement-model fusion approach because correlations are high enough at a sufficient number of sites to make the precipitation adjustment effective at an annual timescale.

Since the focus of this manuscript is regional trends from EQUATES, we feel annual values are valid in this application. However, to guide readers to seasonal analyses from above, we add the following sentence to the methods in Section 2.3:

Since model performance is improved for annual instead of seasonal values (Table S3, refer to Zhang et al. (2019) for detailed seasonal model evaluation), we apply a measurement-model fusion technique previously described by Zhang et al. (2019) to adjust the modeled annual wet deposition fields of inorganic N ($NO_3+NH_4$) and S, briefly described here.

**Specific Comments**

*Abstract* – Overall, the abstract is quite long and I found it challenging to follow. I think it could benefit from a clearer structure.

We thank the reviewer for this feedback. We have revised the abstract to read:

Atmospheric deposition of nitrogen (N) and sulfur (S) compounds from human activity has greatly declined in the United States (US) over the past several decades in response to emission controls set by the Clean Air Act. While many observational studies have investigated spatial and temporal trends of atmospheric deposition, modeling assessments can provide useful information over areas with sparse measurements, although usually have larger horizontal resolutions and are limited by input data availability. In this analysis, we evaluate wet, dry, and total N and S deposition from multiyear simulations within the contiguous US (CONUS). Community Multiscale Air Quality (CMAQ) model estimates from the EPA's Air QUAlity TimE Series (EQUATES) project contain important model updates to atmospheric deposition algorithms compared to previous model data, including the new Surface Tiled Aerosol and Gaseous Exchange (STAGE) bidirectional deposition model which contains land use specific resistance parameterization and improvements to organic N chemistry. First, we evaluate model estimates of wet deposition and ambient concentrations, finding underestimates of $SO_4$, $NO_3$, and $NH_4$ wet deposition compared to National Atmospheric Deposition Program observations and underestimates of $NH_4$ and $SO_4$ and overestimates of $SO_2$ and $TNO_3$ ($HNO_3+NO_3$) compared to the Clean Air Status and Trends (CASTNET) network ambient concentrations. Second, a measurement-model fusion approach employing a precipitation- and bias- correction to wet deposition estimates is found to reduce model bias and improve correlations compared to the unadjusted model values. Model agreement of wet deposition is poor over parts of the West and Northern Rockies, due to errors in precipitation estimates caused by complex terrain and uncertainty in emissions at the relatively coarse 12 km grid resolution used in this study. Next, we assess modeled N and S deposition trends across climatologically consistent regions in the CONUS. Total deposition of N and S in the eastern US is larger than the western US with a steeper decreasing trend from 2002-2017, i.e., total N declined at a rate of approximately –0.30 kg-N/ha/yr in the Northeast and Southeast and by –0.02 kg-N/ha/yr in the Northwest and Southwest. Widespread increases in reduced N deposition across the Upper Midwest, Northern Rockies, and West indicate evolving atmospheric composition due to increased precipitation amounts over some areas, growing agricultural emissions, and regional $NO_x/SO_x$ emission reductions shifting gas-aerosol partitioning; these increases in reduced N deposition are generally masked by the larger decreasing oxidized N trend. We find larger average declining trends of total N and S deposition between 2002-2009 than 2010-2017, suggesting a slowdown of the rate of decline likely in response to smaller emission reductions. Finally, we document changes in the modeled total N and S deposition budgets. The average annual total N deposition budget over the CONUS decreases from 7.8 kg-N/ha in 2002 to 6.3 kg-N/ha in 2017 due to declines in oxidized N deposition from $NO_x$ emission controls. Across the CONUS during the 2002-2017 time period, the average contribution of dry deposition to the total N deposition budget drops from 60% to 52%, whereas wet deposition dominates the S budget rising from 45% to 68%. Our analysis extends upon the literature documenting the growing contribution of reduced N to the total deposition budget, particularly in the Upper Midwest and Northern Rockies, and documents a slowdown of the declining oxidized N deposition trend, which may have consequences on vegetation diversity and productivity.

Line 28: It is unclear how regional NO$_x$/SO$_x$ emission reductions contribute to widespread increases in reduced N deposition. I understand that the reduction in NO$_x$ deposition increases the proportion of N deposited in reduced form, but it seems like this is stating that is contributes to the absolute increase.

We have revised the sentence to read:

> Widespread increases in reduced N deposition across the Upper Midwest, Northern Rockies, and West indicate evolving atmospheric composition due to increased precipitation amounts over some areas, growing agricultural emissions, and regional NO$_x$/SO$_x$ emission reductions shifting gas-aerosol partitioning; these increases in reduced N deposition are generally masked by the larger decreasing oxidized N trend.

*Introduction*
Line 45: This seems like a limited definition of dry deposition, because dry deposition could be deposited on surfaces other than leaves, like soil or water.

We revised the sentence to read:

> After entering the atmosphere, the major nitrogen (N) and sulfur (S) removal pathways occur by precipitation (wet deposition) or uptake by surfaces, such as terrestrial and aquatic vegetation (dry deposition).

Line 57: In my opinion, urban areas and intense agricultural areas are also essential locations that have limited measurements.

We agree, and revise the sentence to read:

> Despite providing critical deposition information, the limited number of NADP and CASTNET sites in essential locations, such as areas with complex terrain, near urban centers, at high elevation, or in forest ecosystems, restrict a thorough understanding on the amount and consequences of deposition.

Lines 75-79: I am confused by the relationship between this project and TDEP, and I would like to see more comparison to TDEP products throughout the manuscript. Is this effort part of TDEP, or will the results here be incorporated into the TDEP products? TDEP products are used extensively by the NADP community, so clarification here would be very helpful.

TDep currently uses an older version of CMAQ and does not employ a measurement-model fusion approach to wet deposition (although these efforts are planned), so a direct comparison would be limited. We have revised text in the introduction to clarify this project and TDEP:

> As such, the NADP Total Deposition Science Committee (TDEP, see www.nadp.slh.wisc.edu/committees/tdep) advances methods to improve estimates of atmospheric deposition from the Community Multiscale Air Quality model (https://www.epa.gov/cmaq) (see Schwede and Lear (2014)). TDEP products only employ a fusion approach to dry deposition and currently use an older version of CMAQ, although efforts to update the model version and incorporate wet deposition fusion described herein are ongoing.

*Methods and Materials* – Throughout the methods and materials, it would be helpful to be extremely clear about the timescale used. It seems like most calculations were done on an annual basis, but this was sometimes confusing.

We have clarified that the timescale used was annual in the text.

Line 121: Was this calculation correcting for chemical transformations performed on an annual basis? Or on a weekly basis?

The revised sentence reads:

> First, modeled wet deposition of NO₃, NH₄, and SO₄ is calculated using the approach by Appel et al. (2011) that accounts for chemical transformations of several species in the aqueous phase. Then, the model and observations are paired in time and space and annually accumulated.

Equation 1: The precipitation correction is done on an annual basis, which seems like it could be problematic. Because N deposition has such a strong seasonal cycle, it matters when the precipitation is either over- or underestimated. If the modeled precipitation is too low mostly during the winter when N concentrations are low, an annual correction could then overestimate N deposition. For more discussion of this problem (and how it introduces error into the NADP annual estimates), see (Schichtel et al., 2019). It would be helpful to see some discussion about the decision to focus on annual values and the issues that this may introduce into the calculations.

We agree that the relationship between wet deposition and precipitation is affected by the frequency, duration, and intensity of the rainfall as well as the ambient concentration. While this relationship is typically nonlinear on hourly and daily time scales, the relationship can become more linear when annually accumulated. The focus on annual values is chosen because NADP maps data at the annual level and the EQUATES simulations covering model years 2002 to 2017 present an opportunity to evaluate modeled regional deposition trends containing important model updates (i.e. consistent emissions methodology, use of STAGE bi-directional model, improve organic nitrate chemistry, etc.). Additionally, end users of EQUATES output for nutrient assessments, including critical loads, are generally more interested in annual instead of seasonal values. While a seasonal evaluation would be interesting, it is beyond the scope of this paper to address.

To address this comment, we add the citation suggested above and add how the relationship between wet deposition and precipitation on annual timescales is more linear:

> While wet deposition relies on the season and precipitation rate (Schichtel et al., 2019), the relationship between precipitation and wet deposition is more linear when annually accumulated.

Line 148: How sensitive is this method to the 300-km radius? How was this radius chosen?

As described in Zhang et al., 2019, the size of the moving window (300 km) was determined by a cross-validation analysis. This was done to dampen any large bias and create a regional estimate of bias.

*Results and Discussion*
Figure 3: Is the precipitation here from PRISM or from NTN rain gauges? I don't think the NTN precipitation depth measurements were mentioned in the sampling method section, so this might be confusing to people who are unaware that NTN measures precipitation depth.

The precipitation plotted in Figure 3 is from NTN. The figure is labeled "NADP NTN Precipitation (cm)" and we revise the figure legend to clarify the modeled precipitation is being compared to NADP NTN precipitation observations:

> Figure 1. Scatter plots of annual accumulated bias-adjusted modeled and NTN observed wet deposition (kg/ha) of ammonium (a, NH₄), nitrate (b, NO₃), and sulfate (c, SO₄) from 2002 to 2017 colored by the climate region. Panel d shows NADP NTN observed and modeled precipitation (cm).

Lines 206-224: I found it confusing that this paragraph mixes general results on spatial variability in total deposition with model performance.

This part of the manuscript is discussing model evaluation, including the different spatial regional descriptions.

Lines 264-267: How are you defining hotspots here, and how were they identified? There are many urban areas that are known hotspots of N deposition (e.g., Denver-Boulder metro area), but these do not appear on the map in Figure 5 (but perhaps this is because of the spatial or color scale?).

The color scale and color map does not allow for easy viewing of urban centers since the focus of this manuscript was regional trends from 2002 to 2017. New maps of 2017 total reduced and oxidized N deposition with adjusted colors indicate urban centers do see large amounts of N deposition. This figure was added to the supplemental information and we add a sentence regarding the larger deposition amounts over urban areas to Section 3.2.1:

> Urban regions in the central and eastern US indicate a substantial amount of N deposition compared to nearby rural areas (Figure S7), consistent with previous findings that bulk N deposition in urban areas is twice as much as rural and remote sites (Decina et al., 2019).

[Figure]

Additionally, we remove the word "hotspot" to avoid confusing readers.

Line 312: Explain the connection between warming temperatures and increasing reduced N deposition more fully. Also, what about on-road emissions of ammonia? These are an increasingly important source of $NH_3$ emissions connected to $NO_x$ emission control mechanisms (Fenn et al., 2018).

The revised text reads:

> Fertilizer use in the Midwest has only grown modestly (~1.3 %/yr), so increases in total N deposition have been largely attributed to increasing reduced N deposition. Growing reduced N deposition is a result of $NH_3$ emissions increasing exponentially with temperature (except below freezing and where emissions are near 0) (Riddick et al., 2016) and by increasing the partitioning of $NH_3$ remaining in the gas phase due to $NO_x$ and $SO_2$ emission reductions (Warner et al., 2017) and increasing importance of on-road mobile emissions of $NH_3$ (Fenn et al., 2018).

Figure 5: It might also be helpful to distinguish between areas with unavailable and not significant trends in Figure 5, because these have very different meanings. In this trend analysis, is it possible to have a significant trend with a slope of zero? Figure 5f appears very white – are these places with very small significant trends, or is the slope actually zero?

Trends are only available where we have PRISM data (e.g. CONUS), so anything outside CONUS is light grey. Alternatively, darker grey indicates where trends are not significant. We have clarified this in the Figure 5 legend:

> Figure 2. Spatial distribution of total N (top) and S (bottom) deposition in 2002 (a and d, kg/ha), 2017 (b and e, kg/ha), and the 2002-2017 annual trend (c and f, kg/ha/yr) with significance at the 95% confidence level. Grey areas in panels (c) and (f) indicate where the trend is unavailable due to lack of PRISM data or not significant (i.e., p-value of the Wald test is greater than 0.05).

The Wald test is testing a null hypothesis that the linear regression slope is 0 with an alternate hypothesis that the slope does not equal zero. Therefore, a significant slope means that is it significantly different from zero (i.e., you cannot have a significant slope of zero).

Figure 6: I am struggling to interpret Figure 6, given the fact that areas without a significant trend are removed. Judging by Figure 5, it seems like this removes the vast majority of many regions. If there is a small decreasing

trend in a corner of a region that generally has had stable N deposition, it seems misleading to represent that as a decreasing trend for the whole region. I'm also confused by what 'data size' refers to in the caption. Is each data point a pixel on Figure 5?

We agree that the non-significant trends should not have been removed. We remade Figures 6, 7, and 8 to set the non-significant trends to 0. Data size refers to the number of CMAQ model grid cells with a significant trend that the average , 5th percentile, and 95th percentile trend is based on. The 5th and 95th percentile lines on each region bar can be used to help assess the range of trends seen for each region.

We have revised the text and figure legends to reflect that all grid cells are included in these figures.

Line 356: Again, I'm curious how you are defining the term 'hotspot.'

We have rephrased this sentence to read:

> Regions of elevated wet and dry reduced N deposition have expanded and increased in magnitude across the CONUS (Figure S10) compared to oxidized N, also observed in the NTN $NH_4$ measurements.

**Technical Corrections**

Line 54: Rephrase so it is clear that the NADP, rather than wet deposition, is the subject of the verb "collecting."

The revised sentence reads:

> Wet deposition, sampled weekly in rain or snow, is measured by the National Atmospheric Deposition Program (NADP) National Trends Network (NTN) since 1978.

Line 174: Define NMB in the text as well as in the figure captions.

NMB is defined on line 70.

Figure 2: It would be helpful to make the dashed and dotted lines more visibly different.

We have edited the figure to use color to make these lines stand apart better.

[Figure]

**References**

Appel, K. W., Foley, K. M., Bash, J. O., Pinder, R. W., Dennis, R. L., Allen, D. J., and Pickering, K. E.: A multi-resolution assessment of the Community Multiscale Air Quality (CMAQ) model v4.7 wet deposition estimates for 2002–2006, Geoscientific Model Development, 4, 357-371, 10.5194/gmd-4-357-2011, 2011.

Bettez, N. D., Marino, R., Howarth, R. W., and Davidson, E. A.: Roads as nitrogen deposition hot spots, Biogeochemistry, 114, 149-163, 10.1007/s10533-013-9847-z, 2013.

Decina, S. M., Hutyra, L. R., and Templer, P. H.: Hotspots of nitrogen deposition in the world's urban areas: a global data synthesis, Frontiers in Ecology and the Environment, 18, 92-100, 10.1002/fee.2143, 2019.

Fenn, M. E., Bytnerowicz, A., Schilling, S. L., Vallano, D. M., Zavaleta, E. S., Weiss, S. B., Morozumi, C., Geiser, L. H., and Hanks, K.: On-road emissions of ammonia: An underappreciated source of atmospheric nitrogen deposition, Sci Total Environ, 625, 909-919, 10.1016/j.scitotenv.2017.12.313, 2018.

Joyce, E. E., Walters, W. W., Le Roy, E., Clark, S. C., Schiebel, H., and Hastings, M. G.: Highly concentrated atmospheric inorganic nitrogen deposition in an urban, coastal region in the US, Environmental Research Communications, 2, 10.1088/2515-7620/aba637, 2020.

Loughner, C. P., Tzortziou, M., Shroder, S., and Pickering, K. E.: Enhanced dry deposition of nitrogen pollution near coastlines: A case study covering the Chesapeake Bay estuary and Atlantic Ocean coastline, Journal of Geophysical Research: Atmospheres, 121, 14,221-214,238, 10.1002/2016jd025571, 2016.

Rao, P., Hutyra, L. R., Raciti, S. M., and Templer, P. H.: Atmospheric nitrogen inputs and losses along an urbanization gradient from Boston to Harvard Forest, MA, Biogeochemistry, 121, 229-245, 10.1007/s10533-013-9861-1, 2013.

Riddick, S., Ward, D., Hess, P., Mahowald, N., Massad, R., and Holland, E.: Estimate of changes in agricultural terrestrial nitrogen pathways and ammonia emissions from 1850 to present in the Community Earth System Model, Biogeosciences, 13, 3397-3426, 10.5194/bg-13-3397-2016, 2016.

Schichtel, B. A., Gebhart, K. A., Morris, K. H., Cheatham, J. R., Vimont, J., Larson, R. S., and Beachley, G.: Long-term trends of wet inorganic nitrogen deposition in Rocky Mountain National Park: Influence of missing data imputation methods and associated uncertainty, Sci Total Environ, 687, 817-826, 10.1016/j.scitotenv.2019.06.104, 2019.

Schwede, D. B. and Lear, G. G.: A novel hybrid approach for estimating total deposition in the United States, Atmospheric Environment, 92, 207-220, 10.1016/j.atmosenv.2014.04.008, 2014.

Shen, J., Chen, D., Bai, M., Sun, J., Coates, T., Lam, S. K., and Li, Y.: Ammonia deposition in the neighbourhood of an intensive cattle feedlot in Victoria, Australia, Sci Rep, 6, 32793, 10.1038/srep32793, 2016.

Sun, K., Tao, L., Miller, D. J., Pan, D., Golston, L. M., Zondlo, M. A., Griffin, R. J., Wallace, H. W., Leong, Y. J., Yang, M. M., Zhang, Y., Mauzerall, D. L., and Zhu, T.: Vehicle Emissions as an Important Urban Ammonia Source in the United States and China, Environ Sci Technol, 51, 2472-2481, 10.1021/acs.est.6b02805, 2017.

Walker, J. T., Robarge, W. P., and Austin, R.: Modeling of ammonia dry deposition to a pocosin landscape downwind of a large poultry facility, Agriculture, Ecosystems & Environment, 185, 161-175, 10.1016/j.agee.2013.10.029, 2014.

Warner, J., Dickerson, R. R., Wei, Z., Strow, L. L., Wang, Y., and Liang, Q.: Increased atmospheric ammonia over the world's major agricultural areas detected from space, Geophys Res Lett, 44, 2875-2884, 10.1002/2016GL072305, 2017.

Zhang, Y., Foley, K. M., Schwede, D. B., Bash, J. O., Pinto, J. P., and Dennis, R. L.: A Measurement-Model Fusion Approach for Improved Wet Deposition Maps and Trends, J Geophys Res Atmos, 124, 4237-4251, 10.1029/2018JD029051, 2019.

---

## Author Comment (AC2)

**Response to reviewer comments is in blue.**

In this study, the authors investigated the long-term trend of deposition of N and S using state-of-the-art regional CMAQ model. Model evaluations for the deposition as well as concentration for specific air pollutants are reasonable. The conclusions are not surprising that the depositions in the US are declining from 2002 to 2107, with contributions of reduced nitrogen increasing and oxidized nitrogen decreasing, which are consistent with several previous studies. In general, this study was well designed and fit into the journal. The authors need to make efforts to improve the reading flow for the manuscript, as well as to improve their quality of figures and tables.

Thank you Dr. Zhang for providing feedback on our manuscript that ultimately resulted in a stronger publication. Specific responses to each comment are provided below.

Change the hyphen "—" to minus "—" through the whole manuscript.

We have made the change as suggested.

**Abstract:**

Line 11-13 "few assess dry deposition, incorporate a measurement-model fusion approach to improve wet deposition estimates, or focus on changes within specific US climate regions." This was exactly what was covered in my two previous studies (Zhang et al., 2018, 2019) which was cited by the authors as well. I suggest the authors refine their motivation or novelty for this study. Also, read the latest paper by Tan et al. (2020) and distinguish the novelty between this study with previous one.

**Reference:**

Tan, J., Fu, J. S. and Seinfeld, J. H.: Ammonia emission abatement does not fully control reduced forms of nitrogen deposition, Proc. Natl. Acad. Sci. U. S. A., 117(18), 9771–9775, doi:10.1073/pnas.1920068117, 2020.

**We have revised the sentence to read:**

While many observational studies have investigated spatial and temporal trends of atmospheric deposition, modeling assessments can provide useful information over areas with sparse measurements, although usually have larger horizontal resolutions and are limited by input data availability.

Line 16—17: Reading from section 2.1, the authors state that the STAGE option was performing similar results as M3dry. So I did not see the point/novelty for the authors to add this statement in the abstract. Also abbreviation for "STAGE" is not necessary since it was not referred again in the abstract.

We add the statement about STAGE in the abstract because this model option was not available until CMAQv5.3. STAGE uses a resistance-based model parameterization (see Massad et al. (2010)) and allows for land-use specific dry deposition estimates, which are important for terrestrial and aquatic ecosystem health applications. We have added the following to the abstract to explain the novelty of STAGE:

Community Multiscale Air Quality (CMAQ) model estimates from the EPA's Air QUAlity TimE Series (EQUATES) project contain important model updates to atmospheric deposition algorithms compared to previous model data, including the new Surface Tiled Aerosol and Gaseous Exchange (STAGE) bidirectional deposition model which contains land use specific resistance parameterization and land use specific deposition estimates needed to estimate the differential impacts of N deposition to different land use types.

**Line 22: Explain "TNO3"**

We have added the definition of TNO3 to this sentence:

First, we evaluate model estimates of wet deposition and ambient concentrations, finding underestimates of SO4, NO3, and NH4 wet deposition compared to National Atmospheric Deposition Program observations and underestimates of NH4 and SO4 and overestimates of SO2 and TNO3 (HNO3+NO3) compared to the Clean Air Status and Trends (CASTNET) network ambient concentrations.

Line 22-23: Is this sentence used to explain the model evaluation of wet deposition, or concentration?

This sentence is referencing the model evaluation of wet deposition. The revised sentence reads:

Model agreement of wet deposition is poor over parts of the West and Northern Rockies, due to errors in precipitation estimates caused by complex terrain and uncertainty in emissions at the relatively coarse 12 km grid resolution used in this study.

Line 27: Will the "increased precipitation" increase both the reduced and oxidized N deposition as well?

Increased precipitation could increase oxidized N deposition, but depends on changes in local concentrations. Analyzing NADP and CAPMoN measurements in the Eastern US and Canada from 1989 to 2016, Feng et al. (2021) found strong correlations between precipitation amount and NH4 wet deposition trends for the Midwest and Mid-Atlantic, with near-zero changes in NH4 wet concentrations. Feng et al. (2021) found slight negative correlations between precipitation amount and NO3 wet deposition in these two regions because, while precipitation is increasing, there are larger decreasing trends in NO3 wet concentrations.

Line 29-30: This is an interesting finding. Can the author provide explanations why this happens?

Emissions of  $NO_x$  and  $SO_2$  have decreased dramatically in response to the regulations set by the Clean Air Act, but the rate of decline has not been constant over the time period considered. Mchale et al. (2021) assessed wet concentration trends at NADP sites in the US, and found larger decreasing trends of wet  $SO_4$  concentrations at 64% of sites during 2000-2017. They note from 2005 to 2010,  $SO_2$  emission decreased at a much faster rate than before 2000. McHale et al. (2021) found similar results for  $NO_3$ , but less strong than  $SO_4$ . We revise the following sentence to explain this finding:

We find larger average declining trends of total N and S between 2002-2009 than 2010-2017, suggesting a slowdown of the rate of decline likely in response to smaller emission reductions.

Line 30: change to "The average annual total N"?

We have revised as suggested.

**Introduction:** Line 69: define "TNO3" and "NHX"

We have defined these terms as suggested.

Line 76: "TDEP" to "TDep"

We have revised as suggested.

Line 97: Please reorganize this sentence. The Hemisphere CMAQ was used to provide BCs for the 12 km CMAQ only, but not used for the data analysis in this study.

We have clarified this sentence. The revision reads:

Lateral boundary conditions for the 12 km grid spacing CONUS domain used in this study were provided by a 108 km grid spacing Northern Hemispheric simulation.

Line 101: "STAGE" was already defined.

We keep the acronym for STAGE here since it is referred to as STAGE in CMAQ documentation.

**Methods**

Section 2.2: Why the authors explain why the criteria for NTN and CASTNET differ with each, "at least 60% annual coverage" for NTN, and "75% annual coverage" for CASTNET?

The 75% annual coverage for CASTNET was chosen to stay consistent with the Referee's CMAQ-based deposition trends study (Zhang et al., 2018). The relaxed annual coverage threshold for NTN (60%) is used to also be consistent with the Referee's measurement-model fusion approach for the wet deposition correction (Zhang et al., 2019). According to Supplemental Section 2 in Zhang et al. (2019):

For the years of this study (2002-2012) there were 261 NADP/NTN sites with at least one year of data that met these four completeness criteria. However only 68 of these sites met the criteria for the full 11-year period, complicating spatial analysis of temporal trends. Here we have relaxed the completeness criteria using a threshold of 60% for Criterion 1, 3, and 4. The Criterion 2 threshold of 90 percent is left unchanged. For calculating trends we used any site that had 10 or 11 years of data that met this CC, resulting in a total of 183 sites included in the analysis. Figure 6 in section 3.4 highlights the impact of the sites that were included after the completeness criteria was relaxed. The additional data help fill in spatial information, while not changing the overall conclusions drawn from the model predicted trends.

Sites are considered for NTN and CASTNET model evaluation if there are observations for at least 13 of the 16 years simulated.

**Results and Discussions**

Line 161: define "ECODEP"

ECODEP is not an acronym.

Line 167: Please provide figure/table for this statement "although the EQUATES precipitation is still biased low on average relative to PRISM."

We have added the following figure to the supplement to support this statement.

**Figure S1.** Left: Scatter plot of annual accumulated precipitation (cm) from PRISM and observed at selected NTN sites, colored by the NOAA climate region. Right: Scatter plot of annual accumulated precipitation (cm) modeled in CMAQ (WRF) and estimated from PRISM, colored by the NOAA climate region. The positive normalized mean bias indicates the PRISM precipitation amounts are larger than the NTN or WRF precipitation amounts.

Line 219-224: Please show a plot/table for this conclusion. Also, discuss the NH4 first and then NO3 and SO4, following the flow of earlier discussions in the same paragraph.

We have rearranged this paragraph to discuss NH4 first, followed by NO3 and SO4. This conclusion was supported by Figure S5 in the original manuscript, copied below:

---

## Author Response (AR2)

The authors responded to each reviewer comment clearly and carefully, and the manuscript is much improved. I think this manuscript will be an important contribution to the atmospheric deposition modeling literature, and I look forward to seeing it in press.

I have only a few very minor wording suggestions:
Line 75: Check subject-verb agreement.

We have edited to reflect proper subject-verb agreement.

Lines 141-142: I appreciate the clarification that the measurement-model fusion was not applied to the dry deposition, but this sentence is now confusing. It is not clear what is not being applied to the dry deposition.

We have broken this sentence into two parts to clarify no adjustment was done to dry deposition:

> The modeled wet deposition fields are adjusted to account for input biases and uncertainty in the chemical and physical processes governing deposition. No corrections are applied to dry deposition due to limited dry deposition measurements.

Line 245: Missing space between SO4 and concentrations.

Thank you for catching this error.

Line 321: The use of "on the other hand" here is confusing because the story remains the same - the trend is still greater in 2002-2009 and then levels out (and becomes slightly positive) in the later time period.

"On the other hand" was used to highlight that the South was the only region that indicated a positive N deposition trend from 2010-2017. This sentence was clarified to read:

> On the other hand, the South is the only region with a very small increasing trend of ~0.01 kg-N/ha/yr of total N deposition with large variability from 2010-2017, but also indicates a larger declining trend of –0.08 kg-N/ha/yr from 2002-2009.

Lines 393-395: The use of "to" instead of a dash to indicate a range of values is confusing here. I would recommend using a dash for each range of values, as is done on line 399 for the percentages.

We have made the change as suggested.